# Structural shifts in China's oil and gas CH$_4$ emissions with implications for mitigation efforts

Junjun Luo[1], Helan Wang[1], Hui Li[1] & Bo Zheng [1,2]✉

Methane (CH$_4$) is a potent and short-lived climate pollutant, with the oil and gas sectors emerging as an important contributor. China exhibited a substantial expansion of oil and gas infrastructures over recent years, but the CH$_4$ emission accounting tends to be incomplete and uncertain. Here, we construct a CH$_4$ emission database of China's oil and gas systems from 1990–2022 with 80% of emissions tracked as refineries, facilities, pipelines, and field sources. Results show that China's oil and gas CH$_4$ emissions have risen from 0.5[0.5–0.6] TgCH$_4$ yr$^{-1}$ in 1990 to 4.0[3.7–4.4] TgCH$_4$ yr$^{-1}$ in 2022, primarily driven by the growing demand for natural gas during the energy transition. The spatial details provided are critical for characterizing emission hotspots, especially in unconventional gas production fields and densely populated eastern regions. This long-time series and spatially explicit CH$_4$ emission database can contribute to informed policy decisions and swift climate action.

Methane (CH$_4$) has been considered the second most important human-influenced greenhouse gas next to carbon dioxide (CO$_2$), with a 28 times greater global warming potential over a 100-year period[1–4]. Due to its shorter atmospheric lifetime, the removal of CH$_4$ is much more rapid than CO$_2$, presenting a key strategy to limit the rate of near-term warming[5]. The Global Methane Pledge proposed by the United States and the European Union in 2021 aims to reduce CH$_4$ emissions by 30% from 2020 levels by 2030[6], helping to achieve the goals of net-zero emissions by 2050 from the Paris Agreement. The oil and natural gas systems are the primary source of CH$_4$ emissions worldwide[7–10], contributing approximately one-quarter of global anthropogenic CH$_4$ emissions of around 360 Tg in 2017[11]. Given the cost-effectiveness of reducing CH$_4$ emissions from these systems[12,13], it has been recognized as one of the best opportunities for combating climate change[14]. The Oil, Gas and Methane Partnership (OGMP), led by the United Nations Environment Programme (UNEP), the European Commission (EC), and the Environmental Defense Council (EDF), has been committed to reducing CH$_4$ emissions of oil and gas systems by 45% before 2025. Thus, there is a need to establish an accurate and detailed CH$_4$ emission inventory of the oil and gas systems, identifying emission trends,

drivers, and hotspots, which is necessary to formulate targeted emission control policies[15].

China is one of the leading global producers and consumers of crude oil and natural gas around the world[16]. However, the national estimated CH$_4$ emissions from the oil and gas systems across existing inventories vary substantially from 1.9 to 3.2 Tg in 2018[17,18], primarily due to inconsistent estimation boundaries and the absence of reliable emission factors[1,16,19]. Based on the bottom-up methods, specific activity data and an emission factor for this activity are adopted to estimate emissions of each life cycle stage of the oil and gas systems from upstream to downstream, aligning with the definitions of IPCC subcategory 1B2[20]. Each life cycle stage is referred to as the "segment" in this study. The massive emission segments contained in the oil and gas systems have led to different and incomplete coverage in previous studies, often lacking detailed emission estimates for all segments[21–27]. Moreover, the country-specific or region-specific emission factors (EFs) remain largely unknown. Field measurements focusing on oil and gas activities are quite limited in China[27], and large discrepancies exist among different measurement techniques[28]. Since the emission factors of the oil and gas systems are

[1]Shenzhen Key Laboratory of Ecological Remediation and Carbon Sequestration, Institute of Environment and Ecology, Tsinghua Shenzhen International Graduate School, Tsinghua University, Shenzhen 518055, China. [2]State Environmental Protection Key Laboratory of Sources and Control of Air Pollution Complex, Beijing 100084, China. ✉e-mail: bozheng@sz.tsinghua.edu.cn

uncertain, most studies derive their emission factors from IPCC guidelines and previous literature[16].

Spatially, previous studies produced $CH_4$ emission maps by allocating emissions that are estimated at country-, province-, or city-level based on gridded spatial proxies (e.g., population or satellite observations of gas flaring)[29,30]. However, such a method cannot represent emission distributions at fine scales in China[17,22,23]. Geographic information on oil and gas facilities is insufficient, possibly resulting in the misallocation or omission of $CH_4$ emitters[31]. Additionally, there is a lack of gridded emission maps for each segment of the oil and gas systems. For example, the Emissions Database for Global Atmospheric Research (EDGAR)[30] provided the $CH_4$ emission map of the oil transport segment in the oil sector, with no further breakdowns for other segments[17]. Gridded emission datasets produced by the Community Emissions Data System (CEDS)[32] did not include segment details. The research gaps mentioned above impede the foundation for investigating emission drivers and spatial hotspots, hindering mitigation policies.

Herein, we fuse multisource information to estimate annual $CH_4$ emissions from China's oil and gas infrastructures for the period 1990–2022 and investigate the individual segment contributions, to analyze emission trends and underlying drivers. We map the emissions of each segment at a spatial resolution of $0.1° × 0.1°$ by integrating and harmonizing an array of geospatial databases to cover relatively complete oil and gas infrastructures in China. Over 80% of national emissions are confined to point/line/field locations where emissions are most likely to occur, which is crucial for illustrating spatial characteristics and temporal changes of $CH_4$ emission hotspots. Please see the Methods section for a detailed description of the methods and datasets used in this study. Furthermore, our national $CH_4$ emission map is evaluated and compared to the Global Fuel Exploitation Inventory (GFEI)[22], an extensively used gridded emission data product, to indicate the impact of using different spatial allocation approaches on the $CH_4$ emissions mapping. The emission dataset we develop can to some extent serve as a reference for $CH_4$ mitigation efforts, targeting key drivers and regions in China's oil and gas sectors, to achieve carbon neutrality by 2060[33].

## Results

### Trends and drivers of $CH_4$ emissions 1990–2022
China's $CH_4$ emissions from the oil and gas systems increased by about 7 times from 0.5[0.5–0.6] $TgCH_4$ yr$^{-1}$ in 1990 to 4.0[3.7–4.4] $TgCH_4$ yr$^{-1}$ in 2022 (Fig. 1a), with an average annual growth rate of 7% yr$^{-1}$ [6%–7%]. Most of the increase occurred after 2000 with a total increase of 3.3 $TgCH_4$ from 2000 to 2022, which accounted for 93% of the total increase during the last three decades. The average annual growth rate after 2000 was 8% per year, double that of the 1990s (4%).

Our estimates of emission trends are broadly consistent with EDGARv8.0[17], China's national inventory reported to the United Nations Framework Convention on Climate Change (UNFCCC)[18], and Liu et al.[21], revealing a slow increase before 2000 and an accelerated rise thereafter (Fig. 1b). However, the magnitude of our emission trends after 2000 (0.15 $TgCH_4$ yr$^{-2}$) is 27%, 17%, and 19% higher than those reported by EDGARv8.0, UNFCCC, and our estimates using IPCC emission factors (the estimates using the same method but utilizing the IPCC default EFs), respectively. This discrepancy is mostly attributed to the marked increase in emissions from the Shaanxi province (Supplementary Figs. 1 and 2) with rapid growth in local gas production (mostly from unconventional gas-tight gas) since the 2000s and high $CH_4$ emission factors of tight gas fields (see Supplementary Discussion 1 for details). Some other bottom-up studies indicate a positive-to-flat emission trend after 2000. For example, after the slight increase (0.11 Tg $CH_4$ yr$^{-2}$) during 2000–2016, CEDSv2021[34] shows zero growth afterward. Generally, our estimates of emission magnitudes are 24% and 42% lower than the estimates of Liu et al.[21] and EDGARv8.0[17],

respectively, before 2010, but close to both inventories after 2010 (Supplementary Fig. 3) and fall within the range of the top-down studies conducted between 2010 and 2019[35–39] (Supplementary Discussion 2 and Supplementary Table 1).

The primary driver leading to the rise after 2000 was the gas sector, increasing by 3.1 $TgCH_4$ and accounting for 95% of the total emission growth during 2000–2022 (Fig. 1a, c). The emission share of gas exploration & production, gas distribution, and gas transmission & storage climbed from 16%, 7%, and 6% in 2000 to 45%, 17%, and 15% in 2022, respectively, making the gas sector surpass the oil sector and become the largest contributor to $CH_4$ emissions of oil and gas systems. In contrast, emissions from the oil sector remained relatively flat from 1990 to 2022. The contribution of oil exploration & production fell from 64% in 2000 to 15% in 2022. This shift in emission patterns corresponds to the continuously growing demand for natural gas in China, driven by the implementation of clean air actions and carbon emission reduction policies, highlighting the crucial bridge role in the energy transition from dirty coal to clean, green natural gas[40].

### Emission distribution and hotspot identification
The $CH_4$ emissions from infrastructures over lands contribute to more than 90% of China's $CH_4$ emissions from the oil and gas systems, estimated at 3.7 $TgCH_4$ yr$^{-1}$ in 2022 (Fig. 2a). The spatial distribution of land emissions illustrates hotspots of the oil and gas facilities (Fig. 2b, c), fields (Fig. 2d), and urban agglomeration (Fig. 2e), which depict the patterns of human and economic activities and their widespread influence on $CH_4$ emissions. Meanwhile, the offshore emissions (0.3 $TgCH_4$ yr$^{-1}$) highlight offshore production platforms and transmission routes that connect coastal cities and offshore production facilities.

China's $CH_4$ emissions are concentrated along the supply chain of oil and gas sectors from production, refinery, storage, and distribution infrastructures, with emission hotspots identified at the key nodes. Field sources (i.e., upstream emissions, including emissions from oil exploration & production, gas exploration & production, and gas processing segments) are the largest $CH_4$ emitter of the oil and gas systems in China, estimated as 2.7 $TgCH_4$ yr$^{-1}$, more than 66% of national $CH_4$ emissions in 2022 (Fig. 2d). Among top 10% of emitting facilities, 76% were onshore fields (Supplementary Fig. 4), which were responsible for 58% of the national total $CH_4$ emissions of the oil and gas systems in 2022, underscoring the necessity to mitigate emissions from these upstream facilities (Supplementary Discussion 3). Northwest China (1.7 $TgCH_4$ yr$^{-1}$, constituting 42% of the national total emissions) and Southwest China (0.6 $TgCH_4$ yr$^{-1}$, 15%) generate the majority of $CH_4$ emissions, with Shaanxi (1.2 $TgCH_4$ yr$^{-1}$), Sichuan (0.4 $TgCH_4$ yr$^{-1}$), and Xinjiang (0.4 $TgCH_4$ yr$^{-1}$) as the top three provincial emitters in 2022 (Supplementary Figs. 1 and 2). This is mainly because the major oil and gas fields (e.g., Ordos, Sichuan, Tarim, and Junggar) are concentrated in these regions (Fig. 2d), which accounted for 19% and 17% of China's oil and gas production respectively in 2022.

The urban distribution sources (i.e., downstream emissions, including emissions from the gas distribution segment) represent the second-largest $CH_4$ source, with an estimate of 0.7 $TgCH_4$ yr$^{-1}$ and contributed to 17% of the totals in 2022 (Fig. 2e) reflecting the dominant role of residential use in gas demand and emissions. East China emitted 0.5 $TgCH_4$ yr$^{-1}$ in 2022, accounting for 12% of the national total. More than 50% of the $CH_4$ emissions in this region come from the urban distribution sources due to the agglomeration of medium and large cities, serving over 150 million natural gas customers in 2022. In East China, the coastal regions such as Shandong (0.2 $TgCH_4$ yr$^{-1}$) and Jiangsu provinces (0.1 $TgCH_4$ yr$^{-1}$) show relatively higher emissions than the other provinces, primarily due to the imported LNG (Fig. 2b). Line sources (emissions from oil transport and gas transmission segments) are estimated at 0.37 $TgCH_4$ yr$^{-1}$ (Fig. 2c), including 0.36 $TgCH_4$ yr$^{-1}$ from gas transmission and 0.01 $TgCH_4$ yr$^{-1}$ from oil transport. The emission distribution pattern of line sources is largely consistent with

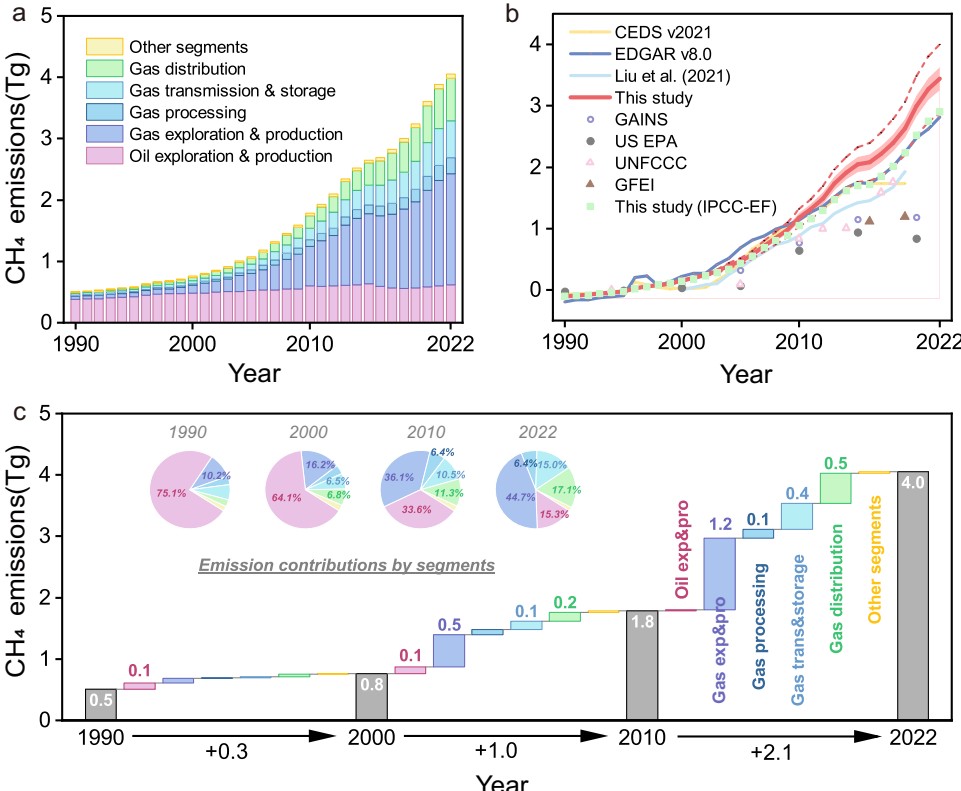

**Fig. 1 | CH₄ emissions from the oil and gas systems in China during 1990–2022.**
**a** Contribution of different segments. **b** Changes in annual CH₄ emissions relative to the average between 1990 and 2000. **c** Driving force analysis. The red-shaded area in (**b**) presents the 95% confidence interval (CI) of our emission estimates. The two red dashed curves in (**b**) present the sensitivity estimates of CH₄ emissions based on the limited and extensive implementation of lower-emitting technologies and practices respectively. This study (IPCC-EF) indicates the estimates using the same method and activity data but applying the IPCC emission factors.

the map of natural gas trunk lines. Sichuan province is the largest contributor to line source emissions (constituting 11% of the emissions from the line sources), owing to the CNPC Sichuan & Chongqing Network's transportation of natural gas.

There is notable spatial variability of the changes in CH₄ emissions from the oil and gas systems after 2000 (Fig. 3). Most production fields in China showed an increasing trend of CH₄ emissions, especially in Northwest, Southwest, and North China, where emissions increased by 1.4 TgCH₄, 0.5 TgCH₄, and 0.4 TgCH₄, respectively, from 2000 to 2022. This corresponds to the surge in domestic production of both conventional and coal bed gas, particularly from the major gas-producing areas such as the Ordos, Sichuan, and Tarim basins. The energy structure and consumption patterns have been shifting in China with the goals of air pollution and climate change mitigation, which stimulates the growing demand for natural gas. The exceptions of the positive trends observed in the producing fields are in Northeast China and Bohai Bay, both of which experienced a continuous decrease in CH₄ emissions over the past 33 years. These decreasing trends are directly related to the reduction in crude oil production volume, possibly driven by increasing extraction challenges and continuously rising oil imports. More importantly, the strategy of "optimizing industrial structure" practiced in the oil-rich provinces has contributed to further declines in oil output. Therefore, major oil fields in China, such as Daqing Oilfield (in Northeast China), are experiencing gradual declines in production, in turn leading to reductions in CH₄ emissions.

The upward emission trend is also evident in urban consumption endpoints, particularly in Eastern China where CH₄ emissions increased by 0.2 TgCH₄ from 2012 to 2022 (Fig. 3i), doubling the increase between 2001 and 2011 (Fig. 3f). This accelerated increase is consistent with the fast development of natural gas distribution

pipeline networks across Eastern China. From 1990 to 2000, CH₄ emissions from gas distribution in Chinese cities were dominated by town gas, a manufactured gaseous fuel produced mainly through the gasification of coal. The composition of town gas differs from that of natural gas, with hydrogen (H₂) being the predominant component, followed by CH₄[20]. On average, town gas accounted for 65% of total emissions from distribution pipelines during this period (Supplementary Fig. 5). Since 2000, the transition of CH₄ emissions from town gas to natural gas in the gas distribution segment has been generally observed across cities (Supplementary Discussion 4). As China is under a rapid phase of urbanization[15], the number of natural gas users has been soaring. Furthermore, China's natural gas utilization policy[41] preferentially ensures urban gas supply, which effectively promotes pipeline construction in cities. The faster growth after 2012 could also be partly explained by the increasing reliance on imported gas, with more than 10 LNG terminals put into operation in eastern port cities like Ningbo, Zhoushan, and Qingdao since 2012.

The expansion of gas transmission pipelines throughout the country has been striking during the last two decades, indicating a speed-up in CH₄ emissions compared to the slower rates observed for the pre-2000 period (Fig. 3a–c). During 2001–2011, the emissions increased by 0.1 TgCH₄ from gas transmission sources. Notably, the West to East gas pipeline (Supplementary Fig. 6) started operations in 2004, delivering gas from the Tarim basin to Eastern China, which is the first major long-distance natural gas pipeline in China. Offshore gas pipelines like Chunxiao-Ningbo were put into operation, successfully utilizing the offshore resources. From 2012 to 2022, the emissions from gas transmission are even more extensive following the implementation of China's Twelfth Five-Year Plan for natural gas development (Fig. 3g–i). Apart from the enhancements of domestic pipelines

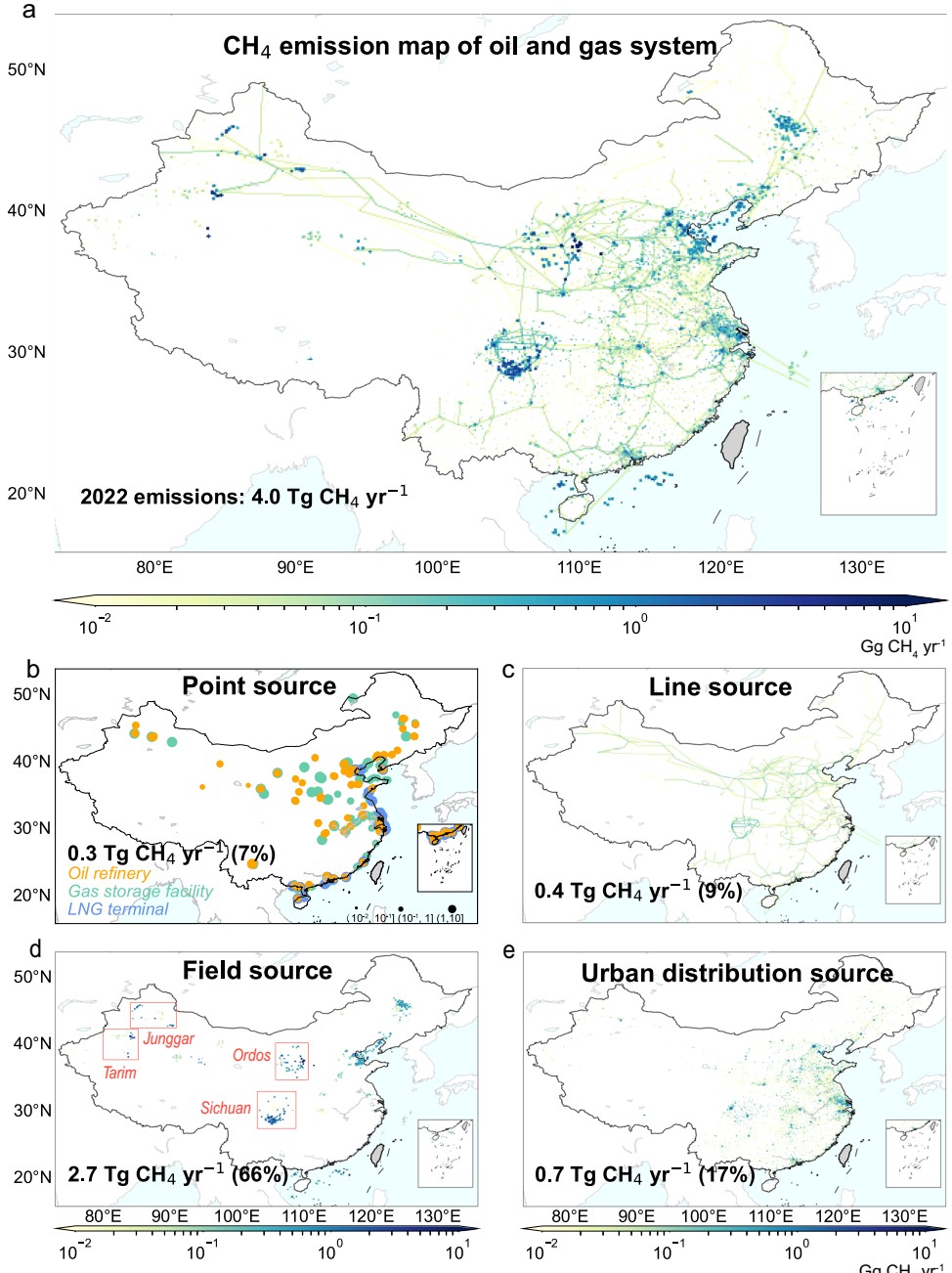

**Fig. 2 | CH₄ emission map of oil and gas systems by source in 2022.** The CH₄ emissions are mapped at 0.1° × 0.1°, including (**a**) total emissions, (**b**) point, (**c**) line, (**d**) field, and (**e**) urban distribution sources. This figure was created using Python 3.9, utilizing the mpl_toolkits.basemap package to import Basemap 1.3.7[58]. No data available for the gray areas on the map.

such as the CNPC Sichuan & Chongqing Network, emissions from imported natural gas pipelines have become evident including the Central Asia Pipeline, the Sino-Myanmar Pipeline, and China–Russia East Pipeline. Currently, China's dependence on imported natural gas has reached 42% and is expected to continue to grow to meet the surging demand. Compared with gas transmission pipelines, CH₄ emissions from the oil transport pipelines have remained considerably lower over the last three decades, by nearly two orders of magnitude, largely due to their lower emission factors.

**Comparisons with existing quantifications**
We evaluate differences in the spatial distribution of gridded emission maps between this work and GFEI[22] (Fig. 4a–c), a gridded emission map

commonly used in inverse analyses. To eliminate the effects of different estimates of emission magnitudes, GFEI emissions were scaled consistently with this work's onshore estimates by segment. The comparison with the GFEI emission map shows that our work identifies emission hotspots in large oil and gas production fields as well as densely populated eastern regions (Fig. 4c). This is mostly due to differences in approaches used for spatial allocation as total emissions from each segment were scaled respectively. The disparities between field sources could be explained by the fact that the upstream emissions were evenly distributed to each upstream facility nationwide by GFEI, without distinguishing between oil and gas, resulting in substantial errors in emissions from large oil and gas production areas. Moreover, the lower estimates of emissions were observed in densely

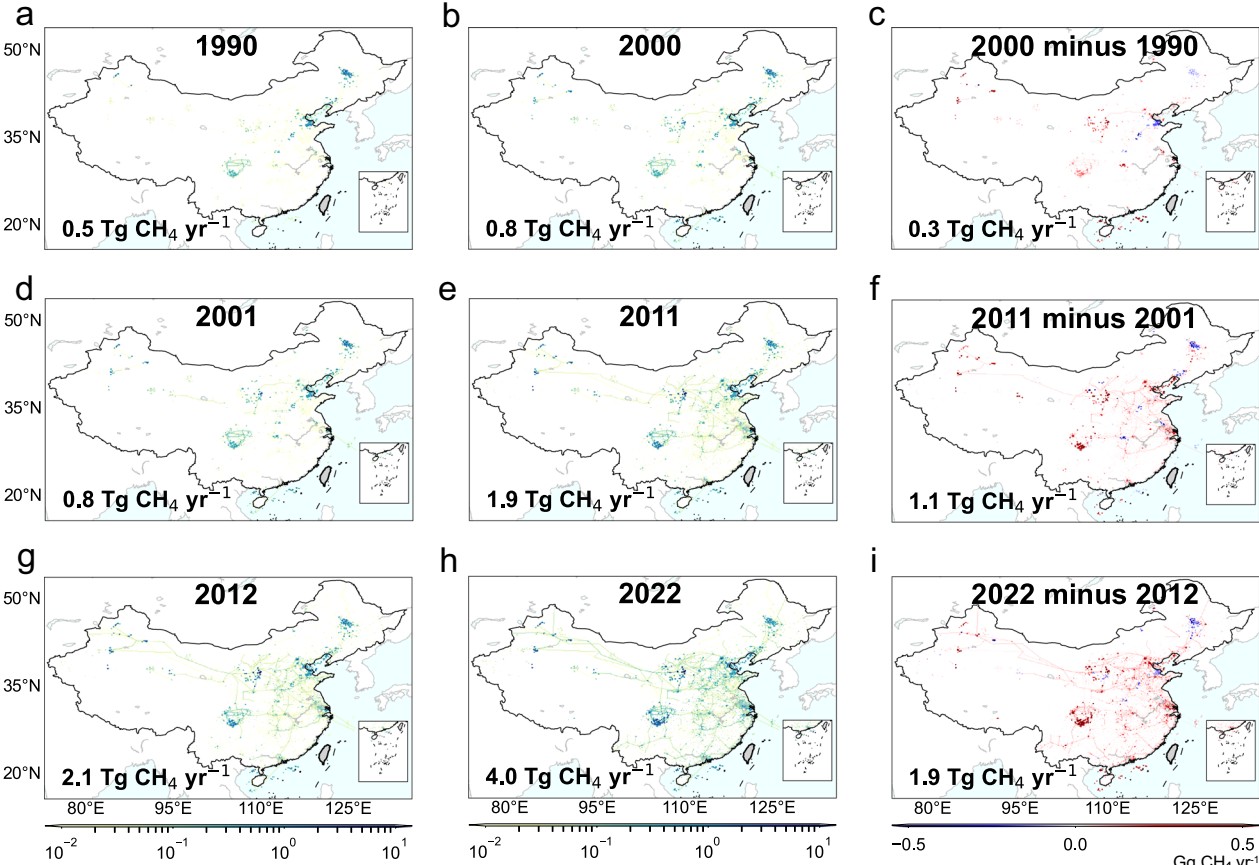

**Fig. 3 | The spatial changes in CH$_4$ emissions from the oil and gas systems in China during 1990–2022.** The CH$_4$ emission maps shown here include the emission distribution at a horizontal resolution of 0.1° × 0.1° in (**a**) 1990, (**b**) 2000, (**d**) 2001, (**e**) 2011, (**g**) 2012, (**h**) 2022, and the spatial distribution of emission differences (**c**) between 1990 and 2000, (**f**) 2001 and 2011, (**i**) 2012 and 2022. This figure was created using Python 3.9, utilizing the mpl_toolkits.basemap package to import Basemap 1.3.7[58]. No data available for the gray areas on the map.

populated eastern regions by GFEI primarily because it allocated gas distribution emissions to grid cells only based on population densities without using an urban land cover map, which may misrepresent emission hotspots, particularly in rural China where the gas distribution pipeline penetration rate remains low[41]. Consequently, the CH$_4$ emission map of GFEI presents a noticeable "smearing" effect (Fig. 4b). The emissions appear to diffuse across a larger area than the actual emission sources, potentially leading to challenges in accurately pinpointing the hotspots of CH$_4$ emissions.

The large spatial discrepancies between the two gridded emission maps are also found in pipelines at 0.1° × 0.1° resolution. The CH$_4$ emissions along pipelines tend to be lower in our work, implying that certain line sources included in GFEI are not represented in our study. After carefully checking each pipeline in both studies, with information on China's oil and gas pipeline infrastructure from Medium- and Long-Term Oil and Gas Pipeline Network Planning[42] developed by the National Development and Reform Commission and National Energy Administration, we identified that some pipelines included in GFEI were either non-existent or inactive in 2019, such as the Tibet Gas Pipeline (Fig. 4c). Emissions were improperly allocated to these pipelines in the GFEI dataset. The reliability of our pipeline information has been evaluated and confirmed by the fact that the total pipeline length well matches the official data acquired from the China Statistical Yearbook[43], after reconstruction and cross-validation of information from multiple sources.

We further carry out a spatial correlation analysis between emissions and population densities at different spatial resolutions for our work and GFEI (Fig. 4d–f). Both studies indicate that the emissions are not proportional to population distribution, as a large fraction of the emissions occur in the oil and gas facilities where few people are located. The most densely populated grid cells that contribute 25% of the total population only account for 11% of oil and gas CH$_4$ emissions in this work and for 7% of emissions in GFEI at 0.1° (Fig. 4d). This result also suggests that the CH$_4$ emissions from densely populated grid cells in this work are higher than those in GFEI, with the maximum cumulative difference reaching 0.1 TgCH$_4$ yr$^{-1}$ from those grid cells with over 400 people km$^{-2}$ (defined as the high population density area) at the spatial resolution of 0.1° (Fig. 4f). The main reason could be that GFEI disaggregated gas distribution emissions to both urban and rural areas based on population density, leading to an underestimation of emissions in urban areas with higher populations.

Furthermore, the discrepancy of emission distributions in high population density areas is spatial resolution-dependent across spatial scales, which decreases substantially from 0.1° to 1.0°. Grid cells with the highest population density that account for 25% of the total population, reflect similar proportions of oil and gas CH$_4$ emissions at 1° in this work and GFEI, reaching 14% and 13% respectively (Fig. 4d). The most densely populated areas contributing to 25% of CH$_4$ emissions in GFEI represent 28% in this work at 0.1°, while it decreases to approximately 25% at 0.5°resolution (Fig. 4e). Coarse-resolution grid cells cover both urban and surrounding rural areas, which can reconcile misrepresentation and allocation errors caused by the proxy-based emission distribution method. These findings suggest that the

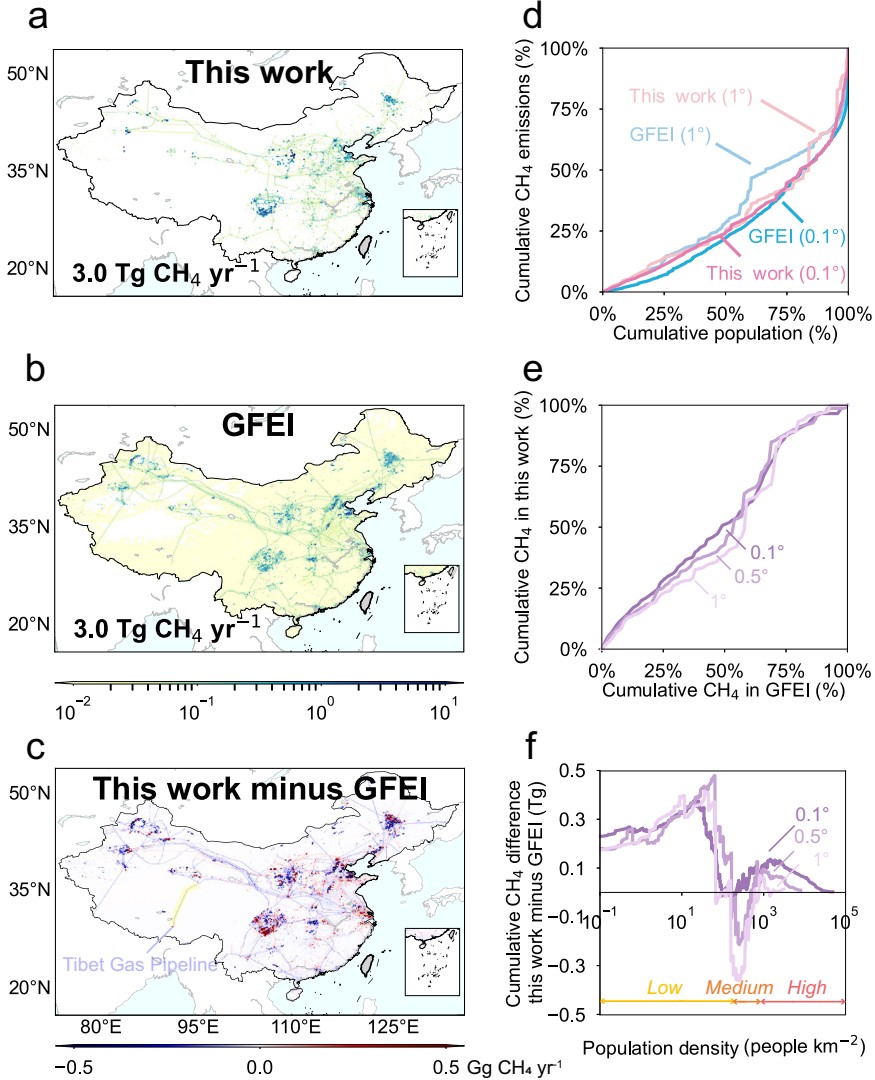

**Fig. 4 | Comparison of gridded CH$_4$ emission maps with GFEI. a** The emission maps of this work and (**b**) GFEI. **c** The differences between this work and GFEI. **d** The comparison of cumulative population and CH$_4$ emissions from this work and GFEI at the spatial resolutions of 0.1° and 1°. The cumulative emissions are calculated based on the descending order of populations. **e** The comparison of cumulative CH$_4$ emissions from this work and GFEI at the spatial resolutions of 0.1°, 0.5°, and 1°.

**f** Differences in cumulative emissions from grid cells with population density larger than one value (y-axes) are plotted along with population density (x-axes) at the spatial resolutions of 0.1°, 0.5°, and 1°. Panels (**a**–**c**) were created using Python 3.9, utilizing the mpl_toolkits.basemap package to import Basemap 1.3.7[58]. No data available for the gray areas on the map.

emission distribution pattern generally agrees with GFEI over coarser spatial resolutions, but the fine-scale emission pattern diverges after adding an urban land cover map for gas distribution emissions in this study.

Spatial distributions of CH$_4$ emissions are also refined in areas with medium (100–400 people km$^{-2}$) and low population density (less than 100 people km$^{-2}$). CH$_4$ emissions from this work are likely to be lower than GFEI in medium population density areas, but higher than GFEI in sparsely populated areas at the horizontal resolutions spanning from 0.1° to 1° (Fig. 4f). This is mainly caused by the allocation of upstream emissions to the grid cells that contain major oil and gas producing facilities. Upstream CH$_4$ emissions of GFEI are distributed based on gridded data that reflects the total number of oil and gas wells per grid cell, without differentiating between oil and gas wells. Given the marked differences in upstream emissions between oil and gas, uniformly distributing total national upstream emissions over all active wells, as GFEI did, could introduce considerable errors in large upstream infrastructure located in medium or low-population-density areas. By contrast, this study separately allocated upstream CH$_4$

emissions from onshore oil, offshore oil, onshore gas, and offshore gas sources to their central locations of the corresponding type of producing fields. We also investigated the areas of typical oil and gas fields in China and created a buffer zone around each point location (see Methods) to mitigate the overestimations of emissions from production fields with the center-point-based emission allocation approach. Thus, the distribution pattern of CH$_4$ emissions in both moderately and sparsely populated areas has been improved with the updated buffer-zone-based method.

## Uncertainties and limitations

Although our national CH$_4$ emission inventory of the oil and gas systems generally agrees with related published studies, indicating the accounting methods adopted are reliable and our results are reasonable, there are still uncertainties and limitations in emission estimation and mapping. First, the uncertainty ranges of the estimated results during 1990–2022 are approximately 5%, which is mainly caused by the quality of activity data and emission factors used. The errors in the activity data primarily arise from the statistical errors during data

collection and release, as the activity data used in the calculation model are mainly derived from statistical yearbooks. We fuse, calibrate, and cross-validate multisource data to ensure the accuracy of activity data in our inventory as much as possible. For example, the field-level production is under the constraint of total onshore production of each province or even city. However, due to the challenge of gathering all activity data associated with the oil and gas systems every year and in every city, we have complemented the missing data based on the proportion of a city's data relative to the country's total in adjacent years. Second, the use of emission factors could introduce additional errors.

Given the lack of temporally varying emission factors for China, we currently apply fixed emission factors from the existing literature[44-46], and IPCC guidelines[20] without conducting field measurements. We focused on trend analysis in this study, considering that the uncertainties in emission factors may cause systematic errors that could be substantially reduced when analyzing trends. To address the concerns related to the use of fixed factors, we conducted a sensitivity analysis based on the assumptions outlined in the GAINS and USEPA frameworks for non-$CO_2$ emissions[47]. Given that China is classified as an emerging economy, we recognize that the abatement technologies in China are not as mature as those in developed countries. We have assumed that between 2000 and 2022, mitigation measures in China led to a 1% annual reduction in upstream emission factors for oil and gas systems. For midstream and downstream emissions, such as pipeline leaks, there have been no corresponding reduction policies implemented, and related technologies remain underutilized. For the period from 1990 to 2000, we did not account for improvements in emission reduction technologies, as this was a foundational phase for the industry. The results of our sensitivity tests indicate that our emission trends exhibit robustness, with national emission variations not exceeding 16% from 2000 to 2022 (Supplementary Fig. 7). We also conducted a sensitivity analysis to investigate the influences of using different IPCC emission factors (see Methods). In the future, more measurements of emission factors for China's oil and gas systems would help improve emission monitoring and evaluation.

The urban distribution pipeline systems are usually complicated, making it harder to accurately pinpoint the natural gas release[48]. $CH_4$ emissions from these sources have not yet been regarded as line sources in this work, which are disaggregated from cities to grid cells using gridded proxies, assuming emission intensities increase proportionately with urban population densities. This is a reasonable way to address gas distribution emissions, as urban populations are the primary consumers of gas, and the construction of distribution pipes is directly related to the number of gas customers. Additionally, our $CH_4$ emission map rectifies the previous errors of emissions from upstream facilities with the uniform emission allocation approach, based on the locations of center points of the oil and gas fields, their respective field types, and the consideration of buffer zone shapes. Although it is sensible to a certain extent, we still fail to represent emission distribution patterns of the upstream emissions at the well-scale. A geospatial database of oil and gas infrastructure—including urban distribution pipelines, and onshore and offshore oil-gas well locations—should be incorporated to support high-resolution $CH_4$ emission mapping for China in the future.

## Discussion

There is substantial spatial heterogeneity in emission changes in China from 1990 to 2022. $CH_4$ emissions of the oil and gas systems in oil-dominated provinces, such as Heilongjiang and Shandong, have decreased or remained stable (Supplementary Fig. 2). In contrast, most provinces in China have shown positive trends in $CH_4$ emissions from the oil and gas systems, largely attributed to the gas sector, reflecting intensified efforts in gas production and gas infrastructure construction in response to national and local energy policies. Currently, $CH_4$

emissions from China's oil and gas industry are highly concentrated within onshore gas fields, particularly in unconventional gas fields (Supplementary Discussion 1), leading to high upstream emissions in western regions of China, such as Shaanxi, Xinjiang, and Sichuan provinces. For example, Shaanxi encompasses China's major tight gas-producing fields that have much higher $CH_4$ emission factors compared to conventional gas fields, reaching 1207.2 Gg$CH_4$ yr$^{-1}$ in 2022 and contributing to 30% of national total emissions.

The primary consumption centers for oil and gas are situated far from these western areas, mainly in densely populated and economically developed regions (provinces with a gross domestic product exceeding 5 trillion yuan in 2022) along the eastern coastline, such as the Yangtze River Delta and Shandong province, which exhibited high downstream emissions but low upstream emissions. In 2022, while 66% of $CH_4$ emissions from the eastern areas were from the urban distribution source, only 18% were from the field source (Supplementary Fig. 8). The geographical disparity between oil and gas fields and major demand centers reveals the production-consumption mismatch of oil and gas resources in China, thereby resulting in the transfer of upstream $CH_4$ emissions from major oil and gas-consuming provinces in the East to the primary production provinces in the West (Supplementary Discussion 5). This contrasting spatial pattern has necessitated the construction of more than 300 long-distance transmission pipelines, which accounted for 9% of the total $CH_4$ emissions from the oil and gas sectors nationwide in 2022.

The $CH_4$ emissions from the oil and gas systems can be substantially mitigated by targeting the key driving processes: gas production, transmission, and distribution (Fig. 1a). Our study demonstrates that more than 70% of the $CH_4$ emissions from the national oil and gas systems come from these segments currently, which are exhibiting rapid growth rates at present. The emission-dominant sector has shifted from the oil sector towards the gas sector since the 2000s with the expanding demand for gas-related products and facilities. Given coal-to-gas switching and the fast urbanization in China, it may be difficult to reduce gas use and production in the near term future, and $CH_4$ emissions from the gas sector are expected to continue to increase. Whether natural gas serves as a feasible transitional fuel for low-carbon economies and societies thus relies upon segment-specific emission control strategies. In such a case, major gas-production fields in Northwest and Southwest China should be given a high priority for emission monitoring and management, including equipment upgrades and leakage-reducing technology deployment. Notably, we suspect that production activities related to unconventional gas could become focal points for China's future mitigation efforts.

The major oil and gas production areas in the West do not receive the same economic benefits from oil and gas resources as the consuming centers in the East, instead, they incur much higher costs associated with emission reduction tied to oil and gas extraction and processing. We suggest that the eastern areas that mostly benefited from oil and gas supplies should invest in emission-reducing technology upgrades in western regions, to facilitate a more equitable distribution of resources and support national $CH_4$ reduction initiatives. Furthermore, Eastern China should promote its mitigation actions regarding natural gas distribution management due to the surge in natural gas end users and the expanding coverage of distribution pipelines. Given the long-standing imbalance between production and consumption in China's oil and gas industry, the continued construction of additional pipelines may impede the expected effectiveness of $CH_4$ emission mitigation efforts in China, which requires complete identification of high-emission pipelines and preferential application of abatement options along these long-distance routes. In turn, such practices could further obtain credits for energy saving and selling power, leading to both economic and environmental co-benefits.

**Table 1 | Data sources for emission estimation in this study**

| Emission segment | Emission source | Activity data | Data source | Incoming resolution |
|---|---|---|---|---|
| Oil exploration | Oil producing fields | Oil production volume | China's National Bureau of Statistics (https://www.stats.gov.cn/), China Land & Resources Almanac[51], City Statistical Yearbooks[52] | Province/City |
| Oil production | | | | |
| Oil transport | Oil transport pipelines | Oil transport volume | China's National Bureau of Statistics | Country |
| Oil refining | Oil refineries | Oil refining volume | China's National Bureau of Statistics | Province |
| Gas exploration | Gas producing fields | Gas production volume | China's National Bureau of Statistics, China Land & Resources Almanac[51], City Statistical Yearbooks[52] | Province/City |
| Gas production | | | | |
| Gas processing | | | | |
| Gas transmission | Gas transmission pipelines | Length of transmission pipeline | China Statistical Yearbook[43], Natural Gas Development Report[53], Medium - and Long-Term Oil and Gas Pipeline Network Planning[42] | Pipeline |
| Gas storage | Gas storage facilities | Gas consumption volume | China's National Bureau of Statistics | Province |
| Gas import/export | LNG facilities | Number of LNG station | Global Energy Monitor (https://globalenergymonitor.org/) | Point |
| Gas distribution | Gas distribution pipelines | Length of distribution pipeline | China Urban Construction Statistical Yearbook[55] | City |

The information includes emission segments, emission sources, activity data, data sources, and spatial resolutions of $CH_4$ emissions in the study.

The detailed emission map constructed from point, line, and field sources in this study has the potential to enhance $CH_4$ emission monitoring and inversion in China. The current techniques that infer local $CH_4$ emission sources from observations rely on high-resolution emission maps to build a connection between $CH_4$ emission sources and concentrations sampled by atmospheric measurement. Gridded emission maps established previously have considerable uncertainties at fine scales and could introduce large errors in inverse modeling. We improve China's $CH_4$ emission distribution patterns of the oil and gas systems at a horizontal resolution of 0.1°, which represents the spatial characteristics adequately. This emission inventory data product improves the emission mapping accuracy over different population density areas and can increase the possibility of observing $CH_4$ emission plumes through satellite sensors from space. With new satellites providing us with ultra-high resolution satellite imagery in the future[49,50], we need more spatially explicit $CH_4$ emission inventories to interpret $CH_4$ enhancements, quantify source-background gradients, and relate them with $CH_4$ emission sources nearby to support emission monitoring.

## Methods

### General framework

We develop a 0.1° × 0.1° gridded inventory of $CH_4$ emissions from China's oil and gas systems from 1990 to 2022. Our framework covers eleven major segments of the oil and gas systems. The first step of our work is calculating the $CH_4$ emissions for each segment. The second step refers to spatially allocating $CH_4$ emissions from each segment to 0.1° × 0.1° grid cells. With regarding different mapping approaches, these emission segments are further classified into four categories, point sources, line sources, field sources, and other sources (i.e., the urban distribution source in Fig. 2). Finally, the $CH_4$ emission maps of different segments are aggregated to form our final emission map.

### Emission estimation

We estimate the annual $CH_4$ emissions of the oil and gas systems using the following equation:

$$E_t = \sum_k A_{k,t} \times EF_k \tag{1}$$

where $t$ and $k$ represent year and emission segment respectively. $E$ represents $CH_4$ emissions from the oil and gas systems (metric tonnes), $A$ represents specific annual activity data (units such as million

cubic meters onshore conventional gas produced), and $EF$ represents the emission factors that are derived from China's official inventory[46], the IPCC guidelines[20], and other literature[44,45] (tonnes per unit of activity). Our inventory integrates multi-source data to develop a more granular and complete set of activity data (Supplementary Tables 2 and 3), along with field-specific emission factors for China.

We incorporate local emission factors wherever possible based on a comprehensive review of existing studies on emissions of China's oil and gas industry (Supplementary Tables 4-6). In cases where the local factors are not available, the Tier 1 emission factors recommended by the IPCC are adopted instead. More than 65% of total emissions are calculated based on China's emission factors. We also use the same calculation method and the same activity data but adopt the IPCC default EFs to evaluate the influence of the local EFs in this study on the emission estimates. The types of technologies and practices, such as leak detection and repair (LDAR) programs, can greatly affect emission factors. Two sets of emission factors are provided in IPCC, with the higher set representing limited use of lower-emitting technologies and practices, and the lower set representing extensive use of these methods. However, the extent of implementation of such low-emitting technologies and practices remains unclear in China, so we take the average EF based on both sets and conduct a sensitivity analysis to investigate the influences of directly using high and low IPCC EF values on emission estimates. The details of each emission segment are provided in Table 1. Further explanations for the estimates of upstream emissions from the oil and gas systems, gas transmission, and distribution segments are described in the following paragraphs. The calculation of other segment emissions is relatively straightforward, which can be estimated with activity data listed in Table 1 and EF from China's official inventory.

For oil exploration and production, oil production volume is used as activity data. The annual production is firstly constrained by province based on China's National Bureau of Statistics (https://www.stats.gov.cn/), which provides the total oil production volume in each province without differentiating onshore and offshore production. Then, to separately estimate onshore and offshore $CH_4$ emissions, we derive the offshore oil production in coastal regions including the Bohai Sea, the East China Sea, and the South China Sea from the China Land and Resources Almanac[51]. Third, city statistical yearbooks[52] are integrated into the provincial framework to supplement, calibrate, and correct onshore oil production by city. Specifically, we compile onshore oil production data for 49 prefecture-level cities, where city-level production accounts for more than 90% of the national totals every year.

Province/city-level emission factors of oil exploration and production are assessed using field-level emission intensity data in 2015 from Masnadi et al.[45]. Greenhouse gas (GHG) emission intensities and daily production of 9 representative onshore oil fields, which produced approximately 30% of China's onshore crude oil production volume in 2015, are provided. We identify the provinces or cities locating these oil fields and secondarily calculate the volume-weighted average carbon intensity of each province/city that corresponds to the scale of activity data. $CH_4$ emission factors of each province/city are estimated by taking the national average non-$CO_2$ emission ratio, with the assumption that all non-$CO_2$ emissions consist of $CH_4$. For certain provinces or cities whose field-level emission intensities are not included in Masnadi et al, parameters of adjacent areas are used since we assume that the field structure and extraction technology are comparable. Similarly, $CH_4$ emissions factors for three maritime regions are estimated based on GHG emissions intensities of 5 representative offshore fields. Consequently, onshore and offshore oil emissions are separately calculated by multiplying their respective oil production volumes with $CH_4$ emission factors.

For gas exploration, production, and processing, gas production volume is used as activity data. Similar to oil-related methodologies, we differentiate the onshore and offshore gas emissions in each province/city, with the city-level production data contributing on average around 19% to the national total gas production from 1990 to 2022. To provide a more detailed analysis, we further disaggregate onshore gas emissions into conventional gas and unconventional gas (i.e., coal bed gas), with the coalbed methane production derived from China's National Bureau of Statistics. $CH_4$ emission factors for gas exploration, production, and processing at the province, city, or offshore levels are developed by analyzing emission intensities of 59 gas fields in 2016 from Gan et al.[44]. GHG emission intensities for conventional, unconventional, and offshore gas fields, collectively providing over 90% of the national total gas production in 2016, are included in Gan et al. We determine the $CH_4$ emission factors for upstream emissions in the gas sector with a similar method that is used in the oil sector. Our results indicate substantial variability in the province/city-specific $CH_4$ factors of gas exploration and production, ranging from 1.1 to 18.5 g m$^{-3}$, which is necessary to characterize emission hotspots.

Gas transmission emissions are estimated using the length of transmission pipelines as activity data. The country-level length has increased rapidly over the past three decades, expanding from 7950 km in 1990 to 118,000 km in 2022, which is mainly derived from official statistics such as the China Statistical Yearbook[43], China Natural Gas Development Report[53], and Medium- and Long-Term Oil and Gas Pipeline Network Planning[42]. Pipeline-level length is sourced from the Global Energy Monitor (GEM) (https://globalenergymonitor.org/), which provides information on transmission routes, current status, and start year of operation. First, to calculate the length of each pipeline, we set the projection of CGCS2000_3_Degree_GK_CM_111E, which is fit for the Chinese region as it minimizes deformation and well preserves distance estimates. Upon examination of the total pipeline length with the country-level statistical data for each year, there are evident inconsistencies ranging from 13% to 69% over the past three decades (Supplementary Fig. 9a), partly due to the omission errors of transmission pipelines in the GEM dataset. Additionally, overlapping pipelines in space cannot be fully displayed in a two-dimensional plane in the GEM dataset. For example, the calculated length of the CNPC Sichuan & Chongqing Network from GEM deviates from the official data provided by the China National Petroleum Corporation Limited[54] with discrepancies exceeding 70%. Under the country-level totals, each pipeline's length is evaluated and reconstructed based on official, public, and corporate documents[54]. After calibrating 246 pipelines, the average adjusted length for each pipeline from 1990 to 2022 finally agrees with their recorded data (Supplementary Fig. 9b), enabling us to accurately estimate the gas transmission $CH_4$ emissions.

$CH_4$ emissions from gas distribution are estimated by multiplying the length of distribution pipelines in urban areas by gas-specific $CH_4$ emission factors. Under the constraint of province-level length derived from China's National Bureau of Statistics, this study compiles, harmonizes, and complements the lengths of distribution pipelines for both natural gas and town gas across 347 prefecture-level cities in China from 2000 to 2022 based on the China Urban Construction Statistical Yearbook[55]. Due to the unavailability of urban pipeline length data before 2000, when only national total numbers are available, we use the proportion of city data relative to the country's total in 2000, which is relatively stable from 2000 to 2003, to fill in the missing data.

## Emission mapping

The estimated $CH_4$ emissions for country, province, and city totals are spatially allocated to the oil and gas facilities in space, comprising the oil and gas production fields, crude oil refineries, natural gas storage facilities, LNG facilities, and along the pipelines. These emission entities are divided into 4 categories described in Sect. General framework. $CH_4$ emissions from the point, line, and field sources are mapped based on latitudes and longitudes, transport pipelines, and field areas, respectively, as shown in Fig. 5. The remaining sources of gas distribution emissions are associated with distribution pipeline length at the city scale, where emissions are disaggregated following the distribution of population densities over urban areas.

Point sources are stationary emitting sources that can be located based on their geographical locations[56]. In this study, oil refineries, gas storage facilities, and LNG terminals are key $CH_4$ emission sources for oil refining, gas storage, and LNG import/export, respectively, which are all treated as point sources here. Province-level emissions of oil refining and gas storage are distributed uniformly to the appropriate refineries and gas storage facilities within each province. The geographical locations (latitude and longitude coordinates) of the two types of infrastructure up to 2016 are available from the National Energy and Technology Laboratory's Global Oil & Gas Infrastructure (GOGI) geodatabase (https://edx.netl.doe.gov/dataset/global-oil-gas-features-database). The start year of operation for 176 oil refineries and 307 gas storage facilities in China is unknown and their $CH_4$ emissions contributed only 6% of national totals in 2022, we suppose that their status has remained operational and physical addresses were fixed from 1990 to 2022 (Supplementary Fig. 10). The incoming resolution of the LNG import/export segment is the point source, which is derived from the GEM. It provides the basic information on LNG facilities by 2022, including facility names, unit names, ownership, current status, geographical locations, start year of operation, production capacity, number of stations, and export/import type. Annual emissions from LNG terminals are mapped to the facilities that were operational during each respective year.

$CH_4$ emissions from oil transport and gas transmission mainly occur along the pipelines, which are treated as line sources in this study. Oil transport emissions can occur along railways and pipelines. However, when allocating country-level oil transport emissions, we assume that they only occur along pipelines, since the railway transport routes are not available. It has little effect on the final result, as this segment only accounts for 0.2% of the national total emissions in 2022. We begin by using GEM's pipeline dataset to identify oil transport pipelines that were operational before the end of 2022. Next, we compile the pipelines in service for each year from 1990 to 2022 according to their respective start and end years. $CH_4$ emissions from oil transport are then spatially distributed from the country to grid cells based on oil pipeline length from the GEM. As for the gas transmission emissions, the incoming resolution is the line source. Similar to oil pipelines, the geographical routes of the operating pipelines for each year are collected and the emissions are mapped to these pipelines. The emission distribution at 0.1° is then formed by aggregating

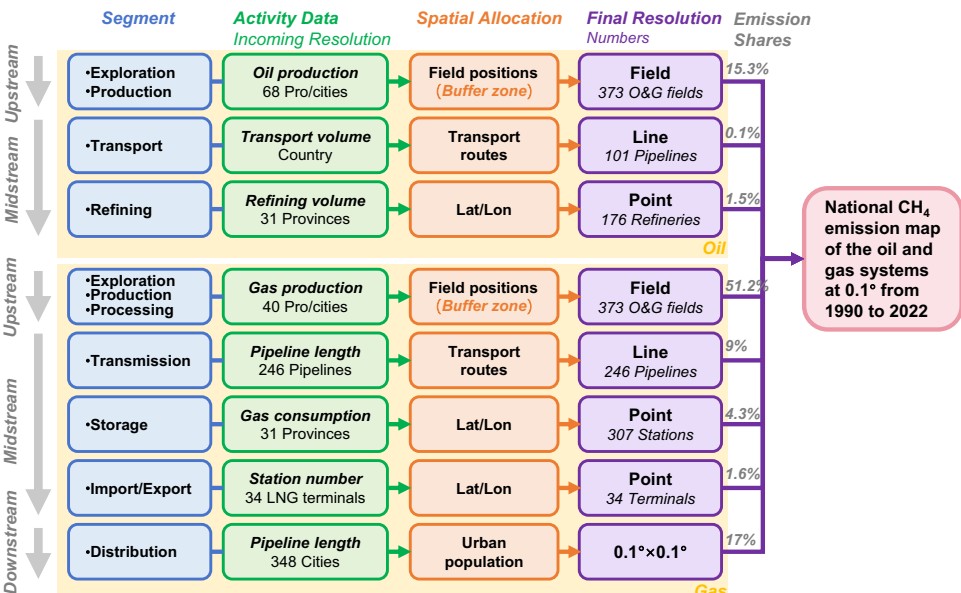

**Fig. 5 | The methodological framework and main data source to develop the CH₄ emission map.** The blue boxes represent different segments of the oil and gas sectors from upstream to downstream, the green boxes indicate activity data, the orange boxes show methods for spatial allocation, the purple boxes represent the final presentation resolution of emissions, and the gray percentages indicate the proportion of emissions in 2022.

the pipeline-level emissions per grid since there may exist multiple pipelines to intersect.

The upstream CH₄ emissions occur in the oil and gas production fields, which are treated as field sources in this study. Information about the production fields is derived from the Havard_OILGAS database (https://hgl.harvard.edu/bookmarks/harvard-glb-oilgas), including field types and locations of center points up to 2016. Due to the unavailability of the field activity status for each year, we assume the activity status depends on whether the city or province where the field is located recorded production volumes during that year. If the production volume was zero, the fields were considered inactive and not emitting. In contrast, if the production volume was greater than zero, the fields were considered active. The collected data from the Havard_OILGAS database could be roughly classified into four types of point locations: onshore oil production, offshore oil production, onshore gas production, and offshore gas production, with a total of 626 point locations. To mitigate the overestimations of emissions from production fields with the center-point-based emission allocation approach, here we investigate the areas of typical oil and gas fields in China and create a buffer zone around each point location as representing the field source. Province/city-level emissions are then allocated to grid cells using the method described below.

First, CH₄ emissions from each province/city are distributed uniformly over the appropriate point locations of oil and gas production within the corresponding province/city. Second, the point data needs to be converted to area emissions. We obtain the mining area of each oil and gas branch company from the China Petrochemical Corporation Yearbook[57] and create buffer zones, circular areas with a 7 km radius around each point, based on the distinct geographic features of oilfields such as Shengli, Henan, and Jiangsu. The overlapping buffer zones are merged into one large shape, with each shape being provided its field name. Third, within the same field, the emissions from points are aggregated to represent the emissions of the field. This database includes 373 oil and gas fields with their names, areas, emissions, and geographical locations. Emissions from each oil and gas production field are downscaled to 0.1° × 0.1° grid cells based on the area of each grid, assuming that emissions are proportional to grid area within each field.

CH₄ emissions of gas distribution are allocated based on total population densities, including both urban and rural populations, in previous research. However, the gas distribution pipe network has not been widely developed in rural areas of China. According to The Natural Gas Utilization Policy, China prioritizes the development of urban gas utilization, indicating that gas distribution emissions mainly occur in urban areas. Therefore, we disaggregate city-level emissions to grid cells based on population densities over urban land areas, using the gridded population map in China from WorldPop (www.worldpop.org), and the gridded land use dataset in China (1980-2015) from the National Tibetan Plateau Data Center (https://data.tpdc.ac.cn/home), with land use cover map provided at five-year intervals. To comply with the resolution of this study, the gridded population (1 km × 1 km) was mapped to the resolution of 0.1° × 0.1°. We then regridded the land use data (1 km × 1 km) to 0.1° × 0.1° and selected the urban area grids (*ugrid*). By integrating land use grid data in 1990, 1995, 2000, 2005, 2010, and 2015 with annual population grids from 2000 to 2022, a total of 23 urban population grids were formed (Supplementary Table 7). The emission mapping process is as follows:

$$E_{ugrid} = \frac{POP_{ugrid}}{\sum_{city} POP_{ugrid}} \times E_{city} \qquad (2)$$

where $E$ represents city-level CH₄ emissions from gas distribution, and $POP$ represents the population.

**Uncertainty assessment**

We used the Monte Carlo method to estimate the uncertainty of our emission estimate. IPCC (2019) provides the uncertainty ranges of emission factors for each segment, which are applied to represent 95% confidence intervals. As for the activity data, production statistics typically have errors of ±15%, and the number of major facilities (e.g., LNG stations) may be relatively accurate[20]. Thus, we suppose the uncertainty range of the number of LNG stations is ±5%, and the uncertainty ranges of other activity data are ±15%. We assume that the uncertainties are symmetric and all parameters, including 16 types of emission factors and 12 types of activity data, are subject to normal

distribution. We performed 10,000 Monte Carlo simulations to estimate the uncertainty of $CH_4$ emission estimates.

## Data availability
The data sources used to construct our high-resolution activity data at the national, provincial, city, and pipeline levels are documented in Supplementary Table 2. The data sources regarding the emission factors in this work are presented in Supplementary Table 4. Field-specific emission intensities utilized to derive local emission factors are presented in Supplementary Tables 5-6. The underlying emission data in this inventory are mainly point, line, and area sources, which are available in their respective formats (https://doi.org/10.6084/m9.figshare.27186936). The annual gridded $CH_4$ emission maps for the oil and gas systems from 1990 to 2022 in China are also available through the same link. Source data are provided with this paper.

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

## Acknowledgements
This work was supported by the National Natural Science Foundation of China (Grant No. 42375096) and the Shenzhen Science and Technology Innovation Commission (Grant Nos. WDZC20220810110301001 and ZDSYS20220606100806014).

## Author contributions
B.Z. conceived and designed the study. J.L. contributed to data collection. J.L. conducted the data analyses and interpretation. The initial draft was prepared by J.L. and B.Z., with contributions from H.L. and H.W.

## Competing interests
The authors declare no competing interests.
