## [Peer Review File · Nature Communications]

Structural shifts in China's oil and gas CH₄ emissions with implications for mitigation efforts

Corresponding Author: Dr Bo Zheng

Version 0:

Reviewer comments:

Reviewer #1

(Remarks to the Author)

Luo et al. constructed an annual CH₄ emission database of China's oil and gas at a 0.1° resolution from 1990-2022 and found that emissions from these systems have increased over this period. I think the key point in this manuscript is not well-suited for Nature Communications, but rather for Earth System Science Data or Scientific Data. The manuscript's perspective in this manuscript lacks innovation, and the content of the manuscript aligns better with the type of data-oriented research.

This manuscript developed a gridded inventory of CH₄ emissions from China's oil and gas by using the bottom-up method. However, no matter the specific annual activity data or emission factor (EF), the manuscript offers little novelty. The EF is mainly from IPCC EF values on emission estimates, which introduces significant uncertainty regarding CH₄ emissions in China. The manuscript mentions that city statistical yearbooks are integrated into the provincial framework to supplement, calibrate, and correct inshore oil and gas production by the city. However, how to account for the mismatch between city-level, provincial-level, and national-level scales?

As is known to all, China experienced an increase in methane emissions from oil and gas in 1990-2022. The increase in methane emissions from oil and gas, rising from approximately 0.7 TgCH₄ yr⁻¹ to 3.8 TgCH₄ yr⁻¹, is relatively small compared to the total methane emissions in China (about 50-70 TgCH₄ yr⁻¹ in Global Carbon Project). The manuscript repeatedly emphasizes the shift in contributions, noting that oil production's share fell from 67% in 2000 to 20% in 2022 and the gas production, gas distribution, and gas transmission and storage increased. However, the increase in natural gas production and the decrease in oil production can be known merely from the data of the National Bureau of Statistics. The percent value is just the sum of oil and gas, which accounts for less than 10% of the total anthropogenic emissions in China. In section 3.2, which primarily discusses the changes in methane emissions from oil and gas, the manuscript analyzes various types of emissions such as field sources, urban distribution sources, and line sources, and their changes across three stages (1990-2000, 2001-2011, 2012-2022). The field source is the largest contributor, followed by changes in urban distribution, with an emphasis on pipeline transportation in China, and the changes also follow this pattern. But, these increases are all related to natural gas, with almost no mention of changes in oil in this part of the discussion. In other words, the increase in methane emissions from natural gas has led to a higher proportion of these emissions compared to oil. This shift is well-known and does not present significant novelty.

The spatial resolution of this inventory is 0.1°, which is insufficient to reflect the distribution of point sources and line sources. China's population is primarily concentrated in the eastern region, resulting in nearly 50% of the CH₄ emissions in this region. Therefore, the spatial details are inadequate for targeting point sources, and almost all emission sources indicate higher emissions in the eastern region. Therefore, I think this manuscript just constructs an emission inventory without providing new insights.

Reviewer #2

(Remarks to the Author)

Luo et al. present a well-executed study compiling a spatially and temporally explicit methane emission inventory for China's oil and gas system. The integration of multi-source data to enhance spatial and temporal representation is valuable for the community to gain a better understanding of China's oil and gas methane emissions and forms a basis for deeper investigation.

The current study still requires improvements to highlight its novelty relative to previous global inventory studies (e.g., Scarpelli et al. 2020 and 2022). Some improvements are shown (e.g., correction of pipeline and oil/gas field data, use urban population as the proxy for natural gas consumption), but they appear incremental.

One unique aspect of this study is the presentation of a long time series from 1990 to 2020, which, as the title indicates, allows for the exploration of structural shifts in China's oil and gas emissions. However, the manuscript currently lacks adequate information on how temporal data is processed, though the spatial mapping information is well presented. For example, some key infrastructure data might be unavailable in early years of the period or not continuous over time (e.g., GOGI). If this is the case, how is the temporal information harmonized? What are the assumptions in generating long-term trends. Including a discussion and clarification on this would be beneficial for readers.

Additionally, it has been empirically shown that emission factors (or emission intensity) from the oil and gas system have been decreasing over time due to technological and management improvements. This aspect is not discussed in the paper. It appears that the authors used fixed emission factors, but changes in EF can have a substantial impact over a long period, which occurs to me a weakness of the

Moreover, I'd request the authors to clarify their plan for data sharing, which is information missing in the current manuscript. The study's value would be greatly enhanced by making the data available to the community.

Minor comments

L138: The calibration of pipeline distribution is well done. It would be worth illustrating this work, for example, by comparing the raw and corrected data in a supplementary figure.

Section 2.3.3: I am not sure if the "buffer zone" method is necessary. It seems GFEI has used well-level information, which should be sufficient for spatially allocating emissions.

Section 2.3.4: It is commendable that the urban-rural difference is accounted for here. Does the urban grid cell mask change over time, or is it fixed?

L47-48: Briefly define what you mean by "segment" here. This is a central concept in the paper, but the meaning is vague in the current description.

L59: There "is a" lack.

L63: didn't -> did not.

L103-104: It may be useful to tabulate the emission factors (EF) used in the study. I also wonder if the study accounts for changes in EF over the long term.

L236: surplus -> surpass.

L238: flattening -> flat/stable.

L224-223: This essentially repeats the last sentence.

L314/L433: "reshaping" usually implies more drastic changes. In this case, it is a bit of an exaggeration, as the general pattern is similar to GFEI. I suggest that "improved" or "refined" better describes the changes

Version 1:

Reviewer comments:

Reviewer #1

(Remarks to the Author)

The manuscript constructs a methane emission database of China's oil and gas system from 1990-2022 using the bottom-up method. Compared with previous global inventory studies, this study extends the time range and harmonizes multi-source data. In the last round of responses, in view of the novelty in research, they gave a detailed reply in terms of the research boundary, data source, estimation method, and policy insights for emissions monitoring of the study. In the long time series of oil and gas methane emissions, it is good to see some novelty phenomena, such as the unexpected significant increase in the Shaanxi natural gas system. Although I mostly think the novelty phenomena can be found directly from activity data and emission factors. I have a few comments for authors to address.

I previously mentioned that this manuscript is more suitable as a data-oriented study. Objectively speaking, the authors have made every effort to collect multi-source data and various emission factors to analyze the correlation between policy implementation and methane emissions from the oil and gas sector. Therefore, the authors emphasized the significance of this work for national policy-making. I partially agree with the manuscript's viewpoint but feel that the importance has been somewhat overstated.

The methane emissions from natural gas in Shaanxi Province (Supplementary Figure 2) were indeed unexpected. In the manuscript, Lines 314-316 mention that this is due to high emission factors. Previously, my understanding was that Sichuan and Xinjiang are the main natural gas production areas. Do other datasets, such as EDGARv8 and GFEL, also show high values for Shaanxi? Could you try to explain the reasons behind the high emission factors in Shaanxi?

Reviewer #2

(Remarks to the Author)

My main concern remains that this long-term study employs fixed emission factors. The authors argue that there have not been dedicated methane reduction efforts in China's oil and gas sector, thus justifying the assumption of a fixed EF. This

argument, however, is flawed. Emission factors can improve through advancements in technology and better management practices, even if these improvements are not specifically aimed at reducing methane emissions. For example, the recovery of associated gas for its economic value can lead to lower emissions.

Empirical global analysis has shown that the average fossil-fuel fugitive emission rate decreased from 7.6% in 1985 to 2.2% in 2013 (Schwietzke et al., 2016). While this FER is not equivalent to EFs used in this study, this shows a long-term improvement in efficiency. If China has followed this global trend, the current study, which uses fixed EFs, may have overestimated the increase in emissions from China's oil and gas sector by a factor of three. This has significant implications for the perceived importance of the sector's emissions. While I understand that obtaining local, time-resolved, segment-specific EFs is challenging, not acknowledging this limitation is problematic for a long-term study.

L173-175: May need to introduce what "town gas" is. I am not familiar with the concept. How do town gas's composition and emission factors differ from pipeline natural gas?

L499-500: the URL is not accessible.

Version 2:

Reviewer comments:

Reviewer #2

(Remarks to the Author)

The authors conducted a sensitivity analysis to evaluate the uncertainty associated with time-varying emission factors, and the findings are presented in Fig. S7. If I understand correctly, Fig. S7 compares the results of "This study (IPCC-EF)" and "This study (time-varying-EF)," and time-varying emission factors are expected to yield lower results compared to fixed emission factors. But based on the figure, the difference between the two (almost overlap) seems smaller than the up to 16% difference mentioned in the text. Moreover, the results for time-varying-EF even appear slightly higher in the figure. Please double-check the accuracy of this figure. Other than this, I have no further comments. The authors have addressed reviewers' comments.

Reviewer: 1

Luo et al. constructed an annual CH₄ emission database of China's oil and gas at a 0.1° resolution from 1990–2022 and found that emissions from these systems have increased over this period. I think the key point in this manuscript is not well-suited for Nature Communications, but rather for Earth System Science Data or Scientific Data. The manuscript's perspective in this manuscript lacks innovation, and the content of the manuscript aligns better with the type of data-oriented research. The manuscript's perspective in this manuscript lacks innovation.

Response:

We extend our appreciation to the referee for valuable feedback on our manuscript, which has greatly assisted our research. Below, we meticulously address each comment in detail.

Compared with previous global inventory studies, this study enhances the completeness of estimation boundaries, harmonizes multi-source data, and refines emission calculation and allocation methodologies to increase the accuracy and details of our inventory. The temporally and spatially explicit annual CH₄ emission database of China's oil and gas systems provided here contributes to capturing province-by-province, city-by-city, and facility-by-facility CH₄ emission trends, which not only benefits data-oriented research but also supports targeted policy-making for emission mitigation in the oil and gas industry. A detailed discussion of these improvements is provided below.

(1) Novelty in research boundary, data source, and estimation method

This research achieves a higher level of completeness in estimation boundaries than previous global inventories, including accounting timescale and emission segments. We constructed a long-time series and up-to-date annual CH₄ emission database for China's oil and gas systems from 1990 to 2022. Due to the challenges associated with intensive data requirements, previous oil and gas studies generally quantified single or discontinuous years, or shorter periods of CH₄ emissions^{1, 2, 3, 4, 5} and cannot analyze emission trends consistently. Moreover, due to the massive emission segments, it is difficult for earlier bottom-up studies to cover complete emission segments (seen in Figure below), often lacking detailed emission estimates for all segments^{1, 2, 3, 4, 5, 6, 7}. This work covers eleven major segments of the oil and gas systems from upstream to downstream and investigates the individual segment contributions, which is crucial for investigating underlying emission drivers and implying CH₄ mitigation in China's oil and gas sectors.

Figure. The emission segments of the oil and gas systems included in existing inventories^{1, 2, 3, 4, 6, 7, 8}.

40 Our inventory integrates multi-source data to develop a more granular and complete set
of activity data, along with field-specific emission factors for China. For a detailed
discussion of the innovations related to activity data and emission factors, please refer
to our response to the first comment below. We also used an array of geospatial
databases to track oil and gas infrastructures in China (seen in the Table below), with
over 80% of national total emissions confined to points, pipelines, and fields where
45 emissions occur.

Table. The geospatial databases for emission mapping in this study.

Emission segment	Emission source	Source type	Geospatial database	Scarpelli et al., 2020 and 2022 ^{2, 3}
Oil exploration & production	Oil and gas field	Field	Havard_OILGAS database (https://maps.princeton.edu/catalog/harvard-glb-oilgas)	
Oil transport	Oil transport pipeline	Line	Global Energy Monitor (GEM) (https://globalenergymonitor.org/)	
Oil refining	Oil refinery	Point	National Energy and Technology Laboratory's Global Oil & Gas Infrastructure (GOGI) geodatabase (https://edx.netl.doe.gov/dataset/global-oil-gas-features-database).	Single data source: GOGI database
Gas exploration & production & processing	Oil and gas field	Field	Havard_OILGAS database	

Gas transmission	Gas transmission pipeline	Line	GEM GOGI database	
Gas storage	Gas storage facility	Point	GOGI database	
Gas import/export	LNG terminal	Point	GEM	
Gas distribution	Gas distribution pipeline	Other	Gridded population map from WorldPop from 2000 to 2020 (www.worldpop.org) Gridded land use map from the National Tibetan Plateau Data Center in 1990, 1995, 2000, 2005, 2010, and 2015 map (https://data.tpdc.ac.cn/home)	Did not use land cover map

Our methods for both emission calculation and spatial allocation have been enhanced. Multi-scale activity data was obtained in this work and was further harmonized across pipeline, city, provincial, and national levels to ensure data consistency. We prioritized national-level data as a top-level total constraint, adjusting smaller-scale data accordingly. A detailed description of the calibration approach is shown in our response to the second comment below. We also refined allocation approaches for CH₄ emissions from both oil and gas fields and urban distribution pipeline systems. By the investigation of typical oil and gas field areas in China and the consideration of buffer zone shapes, the potential overestimations of emissions from production fields were rectified. The distribution pattern of gas distribution emissions was reshaped and improved after adding time-varying urban land cover maps to time-varying population density maps, as urban populations are the primary consumers of natural gas in China.

(2) Novelty in policy insights for emissions monitoring and mitigation

The long-term and high-resolution CH₄ emission inventory of China's oil and gas systems in this study allows us to effectively capture CH₄ emission trends at province, city, and infrastructure levels. Based on this novel dataset, we further analyze the impact of local energy policies on emissions and characterize emission hotspots, which is vital for informing targeted CH₄ reduction strategies in the oil and gas industry. To our knowledge, such a comprehensive analysis that integrates data-driven insights with policy implications for mitigation, at such a fine spatial scale and over a long period, has not been previously achieved in earlier studies. In the following sections (A), (B), and (C), we will provide three examples at the provincial, city, and infrastructure levels to imply focused policy recommendations for mitigation of CH₄ emissions from the oil and gas systems in China.

(A) Province-level perspective

Unexpected increase in emissions from the Shaanxi province

Shaanxi's CH₄ emissions from the oil and gas systems surged more than 400-fold, rising from 2.9 GgCH₄ yr⁻¹ in 1990 to 1207.2 GgCH₄ yr⁻¹ in 2022 (Supplementary Fig. 2).

75 The province's contribution to national total emissions escalated dramatically from 0.7% in 1990 to 30% in 2022. Most of the increase occurred after the 2000s with a total increase of 1154 GgCH₄ between 2000 and 2022, which accounted for 96% of the overall increase over the past three decades. This pronounced growth observed after 2000 coincided with the substantial increase in local gas production, mostly from unconventional gas-tight gas, driven by China's policy to enhance gas production capacity. Shaanxi encompasses China's major tight gas-producing fields, such as Sulige, Daniudi, and Jingbian, which together contribute to more than 90% of the province's total gas production. Notably, these tight gas-producing fields have much higher CH₄ emission factors that can be even ten times higher than those of conventional gas fields⁹.

80

85 The high emission factors of tight gas-producing fields have been considered in the estimation of the Shaanxi-specific emission factor in this study. Therefore, the evident positive-emission trend in Shaanxi is consistent with local-level gas production activities and emission factors.

90 **Supplementary Fig. 2 Variations in CH₄ emissions from the oil and gas sectors for 31 provinces from 1990 to 2022 and the emissions changes in typical provinces.**

However, the unexpected increase in CH₄ emissions in Shaanxi has not been found in any of the previous bottom-up emission inventories for oil and gas systems. This

oversight is because most existing studies estimate emission trends relying on country-level gas production volumes and emission factors, which constrains their ability to reveal distinct emission characteristics of different provinces. In contrast, our bottom-up calculation, with more granular activity data and customized emission factors, effectively analyzes emission changes, identifies emission hotspots, and elucidates underlying causes. We suspect that exploration and production activities related to unconventional gas could become focal points for China's future mitigation efforts.

Contrasting provincial emission trends driven by policy

There is substantial spatial variability in emission changes from 1990 to 2022 in China's 31 provinces (Supplementary Fig. 2). From 1990 to 2022, 26 provinces indicated an increasing trend of CH₄ emissions from the oil and gas systems. Particularly, Shaanxi, Xinjiang, and Sichuan emerged as the highest three provincial emitters in 2022. These positive trends are largely attributed to the gas sector, reflecting intensified gas production efforts in response to national and local energy policies. For example, the accelerated rise in emissions from the gas system over Shaanxi and Xinjiang since the 2000s is largely stimulated by the 10th Five-Year Plan of "enhance natural gas production capacity" and "strengthen gas infrastructure". In Sichuan, CH₄ emissions predominantly arise from the gas sector, with emissions steadily increasing from 55.7 Gg in 1990 to 410.6 Gg in 2022, which is associated with China's 6th Five-Year Plan of preferentially promoting the exploration and production of natural gas resources in Sichuan.

In contrast, CH₄ emissions of the oil and gas systems in oil-dominated provinces, including Heilongjiang and Shandong, have decreased or remained stable. Heilongjiang's CH₄ emissions declined from 153.5 Gg in 1990 to 109.0 Gg in 2022, resulting in a decrease in its percentage of national emissions from 31% in 1990 to 3% in 2022. CH₄ emissions in Shandong remained relatively constant, at around 100 Gg for the whole three decades. These declines or zero trends are directly linked to the reduction in crude oil production volume, possibly driven by increasing extraction challenges and continuously rising oil imports. More importantly, the strategy of "optimizing industrial structure" practiced in these oil-rich provinces has contributed to further declines in oil output. Therefore, major oil fields in China, such as Daqing Oilfield (in Heilongjiang) and Shengli Oilfield (in Shandong) are experiencing gradual declines in production, in turn leading to reductions in CH₄ emissions from the oil system.

Provincial emission transfer embodied in the production-consumption mismatch of oil and gas resources

The emission hotspots have been identified in major gas-producing provinces in the western regions of China, such as Shaanxi, Xinjiang, and Sichuan, rather than in the densely populated and economically developed eastern areas (Supplementary Fig. 7a). This is largely attributed to the remarkable emissions from production-field sources, which are predominantly located in western China. However, the primary consumption centers are situated far from these western areas, mainly in urban regions along the

eastern coastline, indicating the spatial unevenness between production and consumption in China's oil and gas industry. This production-consumption imbalance results in the transfer of upstream CH₄ emissions from major oil and gas-consuming provinces (i.e., downstream provinces) in the east to the primary production provinces (i.e., upstream provinces) in the west. Specifically, in resource-abundant but low-consumption provinces, like Shaanxi, Xinjiang, and Sichuan, upstream CH₄ emissions (i.e., those from field sources) accounted for more than 70% of the total upstream emissions in China, while these provinces contributed only about 10% to the nation's total oil and gas consumption (Supplementary Fig. 7b). In contrast, many coastal and central provinces with limited oil and gas resources show lower upstream emissions but have high consumption for these resources. For instance, in 2022, Shandong accounted for only approximately 2% of national upstream emissions, but its consumption of oil and gas, 164.4 Tg of standard oil (with natural gas converted to crude oil equivalent based on calorific value), surpasses 16% of the national consumption, making it the highest in the country.

Supplementary Fig. 7 The transfer of upstream CH₄ emissions from eastern downstream provinces to western upstream provinces. (a) CH₄ emissions of the oil and gas systems by regions and sources and (b) the comparison of 31 provinces' respective upstream CH₄ emissions relative to national total emissions with their respective oil and gas consumption relative to national total consumption in 2022.

This contrasting spatial distribution of production and consumption has resulted in the construction of more than 300 long-distance transmission pipelines, which contributed to 9% of the total CH₄ emissions from the oil and gas sectors nationwide in 2022. This long-standing imbalance raises concerns that the continued construction of additional pipelines may impede the expected effectiveness of CH₄ emission mitigation efforts in China. Upstream provinces do not receive the same economic benefits from oil and gas resources as downstream provinces, instead, they incur much higher costs associated with emission reduction related to these resources. We suggest that the downstream provinces that mostly benefited from oil and gas supplies should invest in emission-reducing technology upgrades in upstream regions, thereby facilitating a more equitable distribution of resources and supporting national CH₄ reduction initiatives.

Editorial Note: Supplementary Figure 5a-f in this Peer Review File were created using Python 3.9, utilizing the `mpl_toolkits.basemap` package to import Basemap 1.3.7.

(B) City-level perspective

170 CH₄ emissions from the gas distribution segment in China primarily arise from town gas and natural gas. In this study, we obtained the lengths of gas distribution pipelines for both natural gas and town gas across 347 prefecture-level cities, aiming to evaluate the variations in CH₄ emissions from gas distribution sources in detail. Previous bottom-up inventories have not analyzed CH₄ emissions from gas distribution pipelines, over long-term time series at the city level. Furthermore, they have not differentiated between the activity data of natural gas and town gas, despite the different emission factors of these two gas types.

175 From 1990 to 2000, under policies such as the “Seventh Five-Year Plan for Town Gas Development” and the “Technical Policy for Town Gas Development”, CH₄ emissions from gas distribution in Chinese cities were dominated by town gas. During this decade, CH₄ emissions from town gas accounted for 65% of total emissions from distribution pipelines on average (Supplementary Fig. 5). In contrast, CH₄ emissions from natural gas distribution pipelines represented only 35%, and the growth in these emissions was concentrated in a few specific cities, particularly the four municipalities directly under the central government. Specifically, in Beijing, the growth rate of CH₄ emissions from natural gas distribution pipelines during the 1990-2000 period was notable, with an annual increase of approximately 0.2 GgCH₄ yr⁻¹. This rise is largely attributed to the commissioning of the Shaanxi-Beijing natural gas transmission pipeline in 1997, transporting natural gas resources from Shaanxi, Gansu, and Ningxia to Beijing.

190 **Supplementary Fig. 5 The changes in CH₄ emissions from the gas distribution**
segment in China during 1990–2022. The spatial distribution of emission differences
from natural gas distribution (a) between 1990 and 2000, (b) 2001 and 2011, (c) 2012
and 2022, and the spatial distribution of emission differences from town gas distribution
(d) between 1990 and 2000, (e) 2001 and 2011, (f) 2012 and 2022. (g) shows the
195 changes in the proportion of CH₄ emissions from natural gas distribution pipelines
versus town gas distribution pipelines from 1990 to 2022.

The transition of CH₄ emissions from town gas to natural gas in the gas distribution
segment has been generally observed across cities since 2000 (Supplementary Fig. 5).
It is primarily driven by the high costs, poor quality, and environmental pollution issues
200 associated with town gas, leading to its gradual replacement by cleaner and more
affordable natural gas in urban gas supply systems. From 2001~2011, 289 cities
nationwide showed positive trends in CH₄ emissions from natural gas distribution
pipelines. In contrast, 81 cities demonstrated negative trends in CH₄ emissions from
town gas, particularly in eastern coastal metropolitan areas such as Shanghai, Beijing,
205 and Tianjin.

Benefiting from a series of natural gas development plans implemented after the
Twelfth Five-Year Plan, this transition was even more pronounced from 2012 to 2022.
During this period, the increase in CH₄ emissions from natural gas distribution pipelines
in 185 cities was twice that observed from 2001 to 2011. Moreover, in 40 cities, CH₄
210 emissions from town gas distribution pipelines shifted from gains to losses compared
to 2001-2011, mainly located in northern China (Supplementary Figs. 5e and 5f).
Notably, Shanghai experienced robust emission reduction from town gas distribution
during 2012-2022, decreasing emissions by 2 Gg over the ten years and successfully
transitioning to the full utilization of natural gas. Future mitigation actions regarding
215 distribution management should focus on natural gas distribution pipelines, especially
in densely cities in Eastern China.

(C) Infrastructure-level perspective

We assessed infrastructure-specific opportunities for mitigating CH₄ emissions based
on our comprehensive emission dataset which includes 176 oil refineries, 307 gas
220 storage facilities, 34 liquified natural gas (LNG) terminals, 101 oil transport pipelines,
246 gas transmission pipelines, and 373 oil and gas producing fields in China. Previous
global oil and gas inventories have not included such complete geographic information
on oil and gas infrastructures in China^{1, 4, 6, 7, 8}, nor have they spatially distinguished
between oil and gas facilities in their emissions mapping^{2, 3}, which limits their ability
225 to effectively identify emission hotspots. In contrast, our study provides a more detailed
analysis at the level of individual facilities, which is critical for informing policy
decisions and enabling swift climate action targeting the largest CH₄-emitting facilities.

Our findings indicate that targeting a small percentage of high-emitting facilities can
disproportionately reduce CH₄ emissions of the oil and gas systems. Nationally, more
230 than 60% of CH₄ emissions in 2022 from the oil and gas industry were generated by
only 10% of the total oil and gas infrastructure. Among these top 10% of facilities, 76%

Editorial Note: Supplementary Figure 4b in this Peer Review File was created using Python 3.9, utilizing the `mpl_toolkits.basemap` package to import Basemap 1.3.7

were onshore fields, which were responsible for 58% of the national total CH₄ emissions of the oil and gas systems in 2022, underscoring the necessity to mitigate emissions from these upstream facilities. The significance of high-emitting onshore fields is particularly striking in Shaanxi province (Supplementary Fig. 4), where 11 fields (8% of the top 10% of emitting facilities) produced 28% of the national total CH₄ emissions in 2022. It highlights the critical priority of deploying abatement options across the upstream infrastructure in Shaanxi province.

Supplementary Fig. 4 Top 10% of high-emission oil and gas facilities in 2022. (a) shows their CH₄ emissions, and (b) shows the spatial distribution of emissions from these facilities.

Another unique aspect of this study is the more accurate presentation of China's long-distance oil and gas transmission pipeline network for a long period. Under the country-level totals, each pipeline's length is evaluated and reconstructed based on official, public, and corporate documents, enabling the in-depth exploration of the emission expansion from transmission pipelines throughout the country. A speed-up in CH₄ emissions from the line source was observed during the last two decades (Figs. 3d-3i) following the implementation of China's Tenth Five-Year Plan for enhancing natural gas infrastructure. For detailed descriptions of emission growth from onshore, offshore, and imported pipelines, please refer to Emission distribution and hotspot identification in Results. Furthermore, our calibration of pipeline distribution contributed to the comprehensive identification of high-emission pipelines, particularly the CNPC Sichuan & Chongqing Network. The length derived from the GEM dataset is markedly lower than the official data provided by the China National Petroleum Corporation Limited¹⁰, with discrepancies exceeding 70%. By meticulous validation, this study successfully characterized this high-emitting pipeline (Supplementary Figs. 4b and 6), which accounted for 1% of the total CH₄ emissions from the oil and gas sector in China in 2022. With the complete identification of high-emission pipelines, leakage-reducing technologies regarding transmission management can thus be preferentially deployed along these long-distance routes.

1. This manuscript developed a gridded inventory of CH₄ emissions from China's oil

and gas by using the bottom-up method. However, no matter the specific annual activity data or emission factor (EF), the manuscript offers little novelty. The EF is mainly from IPCC EF values on emission estimates, which introduces significant uncertainty regarding CH₄ emissions in China.

Response:

Compared with previous global inventory studies, this work develops a more granular and complete set of activity data, along with reliable emission factors specific to China with multi-source information. A detailed discussion of the improvements in activity data and emission factors is provided below.

(1) Activity data novelty

This study offers comprehensive and up-to-date annual activity data covering a long time series from 1990 to 2022 through the integration, calibration, and cross-validation of multi-source data. The activity data at the national, provincial, city and even pipeline levels were gathered and reconciled to ensure consistency, resulting in the most granular dataset at present to our knowledge (Supplementary Table 3). In addition, the sources of oil and gas have become more diverse, including both onshore and offshore resources, conventional and unconventional (i.e., coal bed gas). Depending on the resource, the emission factors may vary by as much as twofold for production¹¹, which increasingly requires resource-specific activity data. In this study, the oil and gas resources were categorized into offshore and onshore, conventional and unconventional based on a complete and detailed set of activity data (Supplementary Table 2). In contrast, Scarpelli et al. 2020 and 2022 did not employ such detailed classifications.

Supplementary Table 3 The spatial resolution of activity data for different bottom-up inventories.

Activity data	This work	Peng et al., 2016 ¹²	Liu et al., 2021 ¹³	Schwietzke et al., 2014 ⁶	Höglund-Isaksson et al., 2017 ¹	Scarpelli et al., 2020 and 2022 ^{2,3}
Oil production volume	Province/city	Province	Province	Country	Country	Country
Oil transport volume	Country	Unclear	Unclear	Unclear	Not calculated*	Country
Oil refining volume	Province	Unclear	Unclear	Unclear	Not calculated	Country
Gas production volume	Province/city	Province	Province	Country	Country	Country
Length of pipeline transmission pipeline	Pipeline	Not used*	Not used	Not used	Not used	Not used

Gas consumption volume	Province	Not used	Not used	Not used	Unclear	Country
Number of LNG station	Point	Not calculated	Not calculated	Not calculated	Not calculated	Not calculated
Length of distribution pipeline	City	Not used	Not used	Not used	Not used	Not used

290 *Note: Not calculated refers to the exclusion of the corresponding segment from emissions estimation. Not used refers to including this segment for estimation with alternative activity data instead of the activity data presented in the table. For example, the previous studies referenced in this table did not use the length of the transmission pipeline as the activity data to estimate emissions from the gas transmission segment but rather relied on other data, such as the consumption volume. However, according to IPCC guidelines, the length of the transmission pipeline is considered the best indicator of CH₄ emissions from this segment.

295

Supplementary Table 2 The activity data and its multisource information in this study.

Emission segment	Activity data	Data source
Oil exploration & production	Onshore oil production volume	Province-level: China's National Bureau of Statistics (https://www.stats.gov.cn/) City-level: Statistical Yearbook for 50 major oil-producing cities (e.g., Dongying Statistical Yearbook 2022 ¹⁴); Economic Census Yearbook for Guangdong, Henan, and Jiangsu provinces ^{15, 16, 17} ; EPS data platform (https://www.epsnet.com.cn)
	Offshore oil production volume	China Land & Resources Almanac ¹⁸
Oil transport	Oil transport volume	China's National Bureau of Statistics
Oil refining	Oil refining volume	China's National Bureau of Statistics
Gas exploration & production & processing	Onshore conventional and unconventional gas production volume	Province-level: China's National Bureau of Statistics City-level: Statistical Yearbook for 16 major gas-producing cities (e.g., Yulin Statistical Yearbook 2021 ¹⁹); China Economic Yearbook ²⁰ ; Economic Census Yearbook for Jilin, Inner Mongolia, Shanxi, and Sichuan provinces ^{21, 22, 23, 24} ; EPS data platform (https://www.epsnet.com.cn)
		Offshore gas production

	volume	
Gas transmission	Length of transmission pipeline	Country-level: China Statistical Yearbook ²⁵ , Natural Gas Development Report ²⁶ , Medium - and Long-Term Oil and Gas Pipeline Network Planning ²⁷ Pipeline-level: Official, public, and corporate documents for 247 gas transmission pipelines (e.g., Natural gas pipeline transportation cost-related information table ¹⁰)
Gas storage	Gas consumption volume	China's National Bureau of Statistics
Gas import/export	Number of LNG station	Global Energy Monitor (https://globalenergymonitor.org/)
Gas distribution	Length of natural gas and town gas distribution pipeline	Province-level: China's National Bureau of Statistics City-level: China Urban Construction Statistical Yearbook ²⁸

(2) Emission factor novelty

As for emission factors, there is currently a lack of field measurement studies on CH₄ emissions within China's oil and gas industry. The existing measurements are insufficient for establishing a reliable CH₄ inventory covering a long period. The reasons are as follows: (1) The measured CH₄ emission factors across each life cycle stage of the oil and gas systems are still lacking, with most research focusing on the production segment^{29, 30, 31, 32, 33}. (2) The number of measured samples is limited. For instance, Xue et al. conducted measurements at only 18 well sites in the Sichuan shale gas field³⁰. Ge et al. performed field measurements at only 117 wells in the Shanxi Qinshui Basin³¹. (3) There is a lack of long-term continuous measurements, leading to substantial uncertainties in emission trend estimates. Therefore, most of the current CH₄ inventories were constrained to adopt the emission factors provided by the IPCC guidelines^{34, 35, 36} or other literature^{6, 12, 13, 37, 38}.

This study has integrated the most comprehensive and reliable emission factors specific to China to our knowledge (Supplementary Table 4). More than 65% of total emissions were calculated based on China's emission factors. However, previous global inventories often relied on default emission factors from the IPCC guidelines for their emission estimates of each segment. For upstream emissions of the gas and oil sectors, CH₄ emission factors at the province or city levels are developed by analyzing the emission intensities of 14 oil fields and 59 gas fields respectively (Supplementary Tables 5 and 6). The detailed calculation methodology is described in the **Emission estimation** in Methods. Our estimated mean gas-production-normalized CH₄ loss rate of 1.6% during 2010-2017 broadly aligns with satellite-based research conducted nationwide during the same period³⁹, to some extent validating the accuracy of our emission factors.

Supplementary Table 4 Emission factors for different emission segments in this study.

Emission segment	Activity data	Emission factor	Data source
Oil exploration & production	Onshore oil production	Field-specific	Masnadi et al ⁴⁰ .
	Offshore oil production	Field-specific	
Oil transport	Oil transport volume	Country-specific: 0.0061 kg m ⁻³	Official inventory ⁴¹
Oil refining	Oil refining volume	Country-specific: 0.07 kg m ⁻³	
Gas exploration & production & processing	Onshore conventional production	Field-specific	Gan et al ⁹ .
	Coalbed gas production	Field-specific	
	Offshore gas production	Field-specific	
Gas transmission	Length of transmission pipeline	3.09 kg m ⁻¹	
Gas storage	Gas consumption volume	0.48 g m ⁻³	
Gas import/export	Number of LNG station	1660 t station ⁻¹	IPCC, 2019 ¹¹
Gas distribution	Length of natural gas pipeline	0.7 kg m ⁻¹	
	Length of town gas pipeline	0.58 kg m ⁻¹	

325

Supplementary Table 5 CH₄ emission factors for different oil production fields

Field name	Field type	Exploration (kg m ⁻³)	Production (kg m ⁻³)
Huizhou 21-1	Offshore	0.005	24.18
Qinhuangdao 32-6	Offshore	0.002	1.40
Bozhong	Offshore	0.01	5.67
Ansai	Onshore	0.55	1.39
Jingan	Onshore	0.55	1.29
Karamay	Onshore	0.55	1.71
Lamadian	Onshore	0.55	1.49
Saertu	Onshore	0.55	1.59
Take	Onshore	0.56	1.58
Xingshugang	Onshore	0.55	1.65
Jiyuan	Onshore	0.55	1.40
Penglai 19-3	Offshore	0.002	0.87
Suizhong 36-1	Offshore	0.003	0.98
Fengcheng	Onshore	0.65	8.15

Supplementary Table 6 CH₄ emission factors for different gas production fields

Field name	Field type	Exploration &	Processing (g m ⁻³)
------------	------------	---------------	---------------------------------

		Production (g m⁻³)	
Shuangyushi&Jiulongshang	Onshore conventional	0.20	0.97
Anyue	Onshore conventional	0.03	1.05
Datianchi	Onshore conventional	1.06	0.80
Wolonghe	Onshore conventional	0.78	1.19
Mahe	Onshore conventional	1.28	0.89
Kelameili	Onshore conventional	1.65	0.43
Qingshen	Onshore conventional	1.22	0.92
Zhongba	Onshore conventional	1.33	0.86
Sebei	Onshore conventional	1.31	0.82
Tainan	Onshore conventional	1.33	0.82
Kekeya	Onshore conventional	1.33	0.92
Dongping	Onshore conventional	1.23	0.83
Luojiazhai	Onshore conventional	0.00	0.23
Dukouhe	Onshore conventional	1.54	0.87
Longgang	Onshore conventional	0.00	1.51
Tieshanpo	Onshore conventional	0.00	0.55
Hetianhe	Onshore conventional	1.57	1.13
Yuanba	Onshore conventional	4.07	18.19
Puguang	Onshore conventional	1.22	0.00
Dina2	Onshore conventional	2.58	3.97
Kela	Onshore conventional	0.97	0.86
Yingmai7	Onshore conventional	1.29	0.93
Tahe	Onshore conventional	1.02	0.09
Tazhong	Onshore conventional	1.09	0.01
Changling&Songnan	Onshore conventional	0.47	0.00
Sulige	Onshore tight gas	12.06	0.80
Guangan	Onshore tight gas	15.66	0.76
Yingtai	Onshore tight gas	14.75	0.09
Hechuan	Onshore tight gas	13.31	1.44
Yulin	Onshore tight gas	16.56	0.96
Zhaotong	Onshore tight gas	17.25	0.76
Daniudi	Onshore tight gas	16.72	0.79
Bajiaochang	Onshore tight gas	18.64	0.78
Changning&Weiyuan	Onshore shale gas	16.23	0.83
Wushenqi	Onshore tight gas	16.27	1.66
Jingbian	Onshore tight gas	20.06	0.00
Yanchang	Onshore shale gas	19.95	0.30
Mizhi	Onshore tight gas	19.01	0.78
Zizhou	Onshore tight gas	15.95	1.31
Shenmu	Onshore tight gas	20.89	0.75
Xinchang	Onshore tight gas	14.24	0.80
Luodai	Onshore tight gas	17.62	0.81
Fuling	Onshore shale gas	21.39	0.79

Dabei	Onshore tight gas	16.74	0.88
Keshen	Onshore tight gas	17.01	0.89
Juggar CBM	Coalbed methane	4.05	0.83
Qinshui CBM	Coalbed methane	6.71	0.79
Bishuixing CBM	Coalbed methane	7.08	0.83
Ordos CBM	Coalbed methane	10.48	0.91
Chunxiao	Offshore	0.25	0.74
Liwan	Offshore	1.31	0.71
Panyu	Offshore	1.59	0.63
Lingshui	Offshore	1.92	1.46
Bozhong	Offshore	1.80	0.21
Ya	Offshore	2.16	1.17
Wenchang	Offshore	0.45	0.86
Ledong	Offshore	0.00	0.00
Dongfang	Offshore	0.73	1.18
Qiongxi	Offshore	14.76	0.76

2. The manuscript mentions that city statistical yearbooks are integrated into the provincial framework to supplement, calibrate, and correct inshore oil and gas production by the city. However, how to account for the mismatch between city-level, provincial-level, and national-level scales?

Response:

We carefully compared statistical activity data across statistical activity data between city-level, provincial-level, and national-level scales, revealing a high degree of overall consistency, with mismatches averaging around 2% and rarely exceeding 5%. For example, in 2018, the total natural gas production reported by ten gas-producing cities in Xinjiang was 33.57 billion m³, which is only about 4% higher than the provincial figure of 32.19 billion m³ derived from the National Bureau of Statistics.

In this study, the national-level data from the government was prioritized as a top-level total constraint, given that within China’s statistical framework, the national-level data is collected by state statistical agencies through rigorous cross-validation of multiple data sources. In cases where discrepancies arose between national-level data and smaller-scale data, such as provincial or city-level data, we corrected the smaller-scale data to maintain consistency. We first obtained country-level data and then supplemented it with activity data for each province. In a few years where the sum of provincial data did not match the national total exactly, mostly due to minor rounding errors, we reconciled the provincial data based on the national data while maintaining the interprovincial ratios. Similarly, after compiling all available city-level data, if the aggregate of the city-level data did not align with the provincial data, we adjusted the activity data for each city accordingly. This method ensures data coherence across spatial scales and allows us to generate higher-resolution activity data, thereby highlighting the distinct production and emission characteristics of provinces and cities.

3. As is known to all, China experienced an increase in methane emissions from oil and gas in 1990–2022. The increase in methane emissions from oil and gas, rising from approximately 0.7 TgCH₄ yr⁻¹ to 3.8 TgCH₄ yr⁻¹, is relatively small compared to the total methane emissions in China (about 50–70 TgCH₄ yr⁻¹ in Global Carbon Project).

Response:

CH₄ emissions from the oil and gas sectors represent less than 5% of China’s overall CH₄ emissions, however, their contribution to the overall increase in China’s CH₄ emissions from 1990 to 2010 was nearly 10%, according to the Emissions Database for Global Atmospheric Research (EDGAR)⁴². Notably, over the last 10 years, their contribution to the overall increase was even higher, reaching 20%. The surge of emissions in China is mainly encouraged by the current policies such as promoting coal-to-gas transitions, leading to a remarkable increase in China’s demand for natural gas and China has emerged as one of the world’s leading producers and consumers of natural gas¹⁴. Furthermore, the notable growth in CH₄ emissions from the oil and gas sectors is expected to continue or even accelerate, as natural gas represents a key fuel for the 2060 carbon-neutral goal in China. Such a robust upward trend in CH₄ emissions serves as an early warning sign, requiring us to establish an accurate and detailed CH₄ emission inventory of the oil and gas systems to formulate effective and targeted emission control policies timely.

Furthermore, oil and gas CH₄ emissions emerge as one of the most promising opportunities for global climate action⁴³. Firstly, the pathways for reducing them are well-established and cost-effective. According to the International Energy Agency (IEA), approximately 80% of the technologies and practices available for mitigating emissions from the oil and gas systems globally could be utilized at no net cost. By adopting these measures, more than 60% of CH₄ emissions from the oil and gas sectors could be potentially reduced. In addition, CH₄ emissions from the oil and gas industry are highly concentrated, primarily within upstream facilities. For a detailed discussion of high-emitting facilities, please refer to our response to the general comment above. There is substantial potential for CH₄ emission reduction by addressing this relatively small number of infrastructure, buying us time to better manage emissions from other sectors. As a result, both governments and companies have strong incentives to actively pursue emission reductions within the oil and gas industry. Given the high priority for reduction efforts, this work is worth conducting, which offers valuable insights for them to implement low-emitting strategies.

4. The manuscript repeatedly emphasizes the shift in contributions, noting that oil production’s share fell from 67% in 2000 to 20% in 2022 and the gas production, gas distribution, and gas transmission and storage increased. However, the increase in natural gas production and the decrease in oil production can be known merely from the data of the National Bureau of Statistics. The percent value is just the sum of oil and

gas, which accounts for less than 10% of the total anthropogenic emissions in China.

395 **Response:**

The statistical data only provides a macro-level overview and is insufficient for effectively capturing the spatial details of CH₄ emission trends from the oil and gas systems in China. Here, the newly developed long-term and high-resolution emission dataset enables us to thoroughly explore CH₄ emission changes at province, city, and infrastructure levels, which is vital for formulating targeted CH₄ mitigation strategies in the oil and gas systems. To our knowledge, such a comprehensive analysis that integrates data-driven insights with policy implications for mitigation, at such a fine spatial scale and over a long period, has not been previously accomplished before. Detailed analysis at these different levels is shown in our response to the general comment above.

Given that the oil and gas sectors are among the fastest-growing sources of CH₄ emissions in China and a top priority for reduction efforts, this research remains significant. A more detailed explanation can be found in our response to the third comment above.

410

5. In section 3.2, which primarily discusses the changes in methane emissions from oil and gas, the manuscript analyzes various types of emissions such as field sources, urban distribution sources, and line sources, and their changes across three stages (1990–2000, 2001–2011, 2012–2022). The field source is the largest contributor, followed by changes in urban distribution, with an emphasis on pipeline transportation in China, and the changes also follow this pattern. But, these increases are all related to natural gas, with almost no mention of changes in oil in this part of the discussion. In other words, the increase in methane emissions from natural gas has led to a higher proportion of these emissions compared to oil. This shift is well-known and does not present significant novelty.

420

Response:

We have added more discussion of CH₄ emission changes from the oil sector in Line 163-168 and Line 193-195 as follows:

Line 163-168: “These decreasing trends are directly related to the reduction in crude oil production volume, possibly driven by increasing extraction challenges and continuously rising oil imports. More importantly, the strategy of “optimizing industrial structure” practiced in the oil-rich provinces has contributed to further declines in oil output. Therefore, major oil fields in China, such as Daqing Oilfield (in Northeast China), are experiencing gradual declines in production, in turn leading to reductions in CH₄ emissions.”

430

Line 193-195: “Compared with gas transmission pipelines, CH₄ emissions from the oil transport pipelines have remained considerably lower over the last three decades, by nearly two orders of magnitude, largely due to their lower emission factors.”

435 Upon reviewing over twenty existing literature related to CH₄ emissions, the shift of
CH₄ emissions from the oil sector to the gas sector has not been explicitly illustrated
before^{1, 2, 3, 4, 6, 7, 8, 12, 13, 29, 34, 35, 36, 37, 38, 39, 44, 45, 46, 47, 48, 49, 50}. In previous global CH₄
inventories, the discrete annual emission estimates or vague total emissions values fail
to adequately represent the quantitative shift of CH₄ emissions from oil to gas. For
instance, Scarpelli et al. (2020 and 2022) only quantified emissions for 2016 and 2019^{2,}
440 ³. EDGAR only provided total emissions for both sectors without distinguishing their
oil and gas contributions⁷.

Global gridded CH₄ inventories also cannot effectively indicate the spatial shift of CH₄
emissions from oil to gas in China. For example, the Global Fuel Exploitation Inventory
(GFEI), an extensively used gridded emission data product, lacked long-term gridded
445 inventories to pinpoint changes in emission distribution, and GFEI did not differentiate
between oil and gas infrastructure, such as production sites and pipelines^{2, 3}.
Additionally, there is a lack of gridded emission maps for each segment of the oil and
gas systems. For instance, EDGAR only provided the CH₄ emission map of the oil
transport segment in the oil sector, with no further breakdowns for other segments⁷.
450 Gridded emission datasets produced by the Community Emissions Data System (CEDS)
did not include segment details⁴. Regarding domestic CH₄ inventories, although a few
studies indicated that CH₄ emissions from natural gas are gradually surpassing those
from oil, these research lacked high-resolution emission maps and thus primarily
focused on a quantitative shift^{34, 35, 36, 38}. Therefore, the research gaps mentioned above
455 impede the identification of the CH₄ emission shift from the oil sector to the gas sector.

In contrast, this study not only presents the quantitative shift in oil and gas emissions but
also explicitly highlights the spatial shifts across province, city, and infrastructure levels.

460 6. The spatial resolution of this inventory is 0.1°, which is insufficient to reflect the
distribution of point sources and line sources. China's population is primarily
concentrated in the eastern region, resulting in nearly 50% of the CH₄ emissions in this
region. Therefore, the spatial details are inadequate for targeting point sources, and
almost all emission sources indicate higher emissions in the eastern region. Therefore,
I think this manuscript just constructs an emission inventory without providing new
465 insights.

Response:

The underlying data used to build this inventory are mainly point, line, and area sources,
with spatial accuracies greatly surpassing 0.1°. This data will be publicly available in
their respective formats (<https://figshare.com/s/137eaa7d0f63f5d14e6a>) if the paper is
470 accepted. Current CH₄ emission inventories for oil and gas, such as EDGAR and GFEI,
also have a spatial resolution of 0.1°. Besides, our mapping of oil and gas emissions in
China at 0.1° is adequate for use as prior estimates in inverse modeling research.
Existing global inversion model systems typically have resolutions ranging from 1° to
2°, and regional inversion models generally have a resolution of 0.5°.

475 The regions with the highest CH₄ emissions are not the densely populated and
economically developed areas (i.e., the eastern region), but rather the major oil and gas
production areas. Northwest China (1.7 TgCH₄ yr⁻¹, constituting 42% of the national
total emissions) and Southwest China (0.6 TgCH₄ yr⁻¹, 15%) generate the majority of
480 national CH₄ emissions in 2022 (Fig. 2). Our inventory clearly illustrates the transfer
of upstream CH₄ emissions from major oil and gas-consuming provinces in the east to
the primary production provinces in the west. A comprehensive analysis of this
emission transfer pattern is provided in our response to the general comment above.

Reviewer: 2

485 Luo et al. present a well-executed study compiling a spatially and temporally explicit methane emission inventory for China's oil and gas system. The integration of multi-source data to enhance spatial and temporal representation is valuable for the community to gain a better understanding of China's oil and gas methane emissions and forms a basis for deeper investigation.

490 **Response:**

We express our gratitude to the referee for providing positive and constructive feedback on our manuscript. Below, we offer detailed responses addressing each point raised.

1. The current study still requires improvements to highlight its novelty relative to
495 previous global inventory studies (e.g., Scarpelli et al. 2020 and 2022). Some improvements are shown (e.g., correction of pipeline and oil/gas field data, use urban population as the proxy for natural gas consumption), but they appear incremental.

Response:

500 Compared with previous global inventory studies, this study enhances the completeness of estimation boundaries, harmonizes multi-source data, and refines emission calculation and allocation methodologies to increase the accuracy and details of our inventory. A detailed discussion of these improvements is provided below.

(1) Estimation boundary novelty

505 This research achieves a higher level of completeness in calculation boundaries than previous global inventories, including accounting timescale and emission segments. We constructed a long-time series and up-to-date annual CH₄ emission database for China's oil and gas systems from 1990 to 2022. Due to the challenges associated with intensive data requirements, previous oil and gas studies generally quantified single or discontinuous years, or shorter periods of CH₄ emissions^{1, 2, 3, 4, 5} and cannot analyze
510 emission trends effectively. Moreover, due to the massive emission segments, it is difficult for earlier bottom-up studies to cover complete emission segments (seen in Figure below), often lacking detailed emission estimates for all segments^{1, 2, 3, 4, 5, 6, 7}. This work covers eleven major segments of the oil and gas systems from upstream to downstream and investigates the individual segment contributions, which is crucial for
515 explaining underlying emission drivers and exploring structural shifts in China's oil and gas sectors.

Figure. The emission segments of the oil and gas systems included in existing inventories^{1, 2, 3, 4, 6, 7, 8}.

520 (2) Data novelty

Our inventory has fused multisource data to develop a more comprehensive and accurate annual CH₄ emission database of China’s oil and gas systems. The activity data at the national, provincial, city and even pipeline levels were gathered and reconciled to ensure consistency, resulting in the most granular dataset at present to our knowledge (Supplementary Table 3). In addition, the sources of oil and gas have become more diverse, including both onshore and offshore resources, conventional and unconventional (i.e., coal bed gas). Depending on the resource, the emission factors may vary by as much as twofold for production¹¹, which increasingly requires resource-specific activity data. In this study, the oil and gas resources were categorized into offshore and onshore, conventional and unconventional based on a complete and detailed set of activity data (Supplementary Table 2). In contrast, Scarpelli et al. 2020 and 2022 did not employ such detailed classifications.

530

Supplementary Table 3 The spatial resolution of activity data for different bottom-up inventories.

Activity data	This work	Peng et al., 2016 ¹²	Liu et al., 2021 ¹³	Schwietzke et al., 2014 ⁶	Höglund-Isaksson et al., 2017 ¹	Scarpelli et al., 2020 and 2022 ^{2, 3}
Oil production volume	Province/city	Province	Province	Country	Country	Country
Oil transport volume	Country	Unclear	Unclear	Unclear	Not* calculated	Country
Oil refining volume	Province	Unclear	Unclear	Unclear	Not calculated	Country

Gas production volume	Province/city	Province	Province	Country	Country	Country
Length of transmission pipeline	Pipeline	Not used	Not used	Not used	Not used	Not used
Gas consumption volume	Province	Not used	Not used	Not used	Unclear	Country
Number of LNG station	Point	Not calculated	Not calculated	Not calculated	Not calculated	Not calculated
Length of distribution pipeline	City	Not used	Not used	Not used	Not used	Not used

535 *Note: Not calculated refers to the exclusion of the corresponding segment from emissions estimation. Not used refers to including this segment for estimation with alternative activity data instead of the activity data presented in the table. For example, the previous studies referenced in this table did not use the length of the transmission pipeline as the activity data to estimate emissions from the gas transmission segment but rather relied on other data, such as the consumption volume. However, according to IPCC guidelines, the length of the transmission pipeline is considered the best indicator of CH₄ emissions from this segment.

540

Supplementary Table 2 The activity data and its multisource information in this study.

Emission segment	Activity data	Data source
Oil exploration & production	Onshore oil production volume	Province-level: China's National Bureau of Statistics (https://www.stats.gov.cn/) City-level: Statistical Yearbook for 50 major oil-producing cities (e.g., Dongying Statistical Yearbook 2022 ¹⁴); Economic Census Yearbook for Guangdong, Henan, and Jiangsu provinces ^{15, 16, 17} ; EPS data platform (https://www.epsnet.com.cn)
	Offshore oil production volume	China Land & Resources Almanac ¹⁸
Oil transport	Oil transport volume	China's National Bureau of Statistics
Oil refining	Oil refining volume	China's National Bureau of Statistics
Gas exploration & production processing	Onshore conventional and unconventional gas production volume	Province-level: China's National Bureau of Statistics
		City-level: Statistical Yearbook for 16 major gas-producing cities (e.g., Yulin Statistical Yearbook 2021 ¹⁹); China

		Economic Yearbook ²⁰ ; Economic Census Yearbook for Jilin, Inner Mongolia, Shanxi, and Sichuan provinces ^{21, 22, 23, 24} ; EPS data platform (https://www.epsnet.com.cn)
	Offshore gas production volume	China Land & Resources Almanac ¹⁸
Gas transmission	Length of transmission pipeline	Country-level: China Statistical Yearbook ²⁵ , Natural Gas Development Report ²⁶ , Medium- and Long-Term Oil and Gas Pipeline Network Planning ²⁷ Pipeline-level: Official, public, and corporate documents for 247 gas transmission pipelines (e.g., Natural gas pipeline transportation cost-related information table ¹⁰)
Gas storage	Gas consumption volume	China's National Bureau of Statistics
Gas import/export	Number of LNG station	Global Energy Monitor (https://globalenergymonitor.org/)
Gas distribution	Length of natural gas and town gas distribution pipeline	Province-level: China's National Bureau of Statistics City-level: China Urban Construction Statistical Yearbook ²⁸

545 As for emission factors, this study has integrated the most comprehensive and reliable
data specific to China to our knowledge (Supplementary Table 4). More than 65% of
total emissions were calculated based on China's emission factors. For upstream
emissions of the gas and oil sectors, CH₄ emission factors at the province or city levels
are developed by analyzing the emission intensities of 14 oil fields and 59 gas fields
550 respectively (Supplementary Tables 5 and 6). However, previous global inventories
often relied on default emission factors from the IPCC guidelines for their emission
estimates of each segment. We also used an array of geospatial databases to cover
relatively complete oil and gas infrastructures in China (seen in Table below), with over
80% of national emissions confined to point/line/field locations where emissions are
555 most likely to occur.

Supplementary Table 4 Emission factors for different emission segments in this study.

Emission segment	Activity data	Emission factor	Data source
Oil exploration & production	Onshore oil production	Field-specific	Masnadi et al ⁴⁰ .
	Offshore oil production	Field-specific	
Oil transport	Oil transport volume	Country-specific: 0.0061 kg m ⁻³	Official inventory ⁴¹
Oil refining	Oil refining volume	Country-specific: 0.07 kg m ⁻³	

Gas exploration & production & processing	Onshore conventional production	Field-specific	
	Coalbed gas production	Field-specific	Gan et al ⁹ .
	Offshore gas production	Field-specific	
Gas transmission	Length of transmission pipeline	3.09 kg m ⁻¹	
Gas storage	Gas consumption volume	0.48 g m ⁻³	
Gas import/export	Number of LNG station	1660 t station ⁻¹	IPCC, 2019 ¹¹
Gas distribution	Length of natural gas pipeline	0.7 kg m ⁻¹	
	Length of town gas pipeline	0.58 kg m ⁻¹	

Supplementary Table 5 CH₄ emission factors for different oil production fields

Field name	Field type	Exploration (kg m ⁻³)	Production (kg m ⁻³)
Huizhou 21-1	Offshore	0.005	24.18
Qinhuangdao 32-6	Offshore	0.002	1.40
Bozhong	Offshore	0.01	5.67
Ansai	Onshore	0.55	1.39
Jingan	Onshore	0.55	1.29
Karamay	Onshore	0.55	1.71
Lamadian	Onshore	0.55	1.49
Saertu	Onshore	0.55	1.59
Take	Onshore	0.56	1.58
Xingshugang	Onshore	0.55	1.65
Jiyuan	Onshore	0.55	1.40
Penglai 19-3	Offshore	0.002	0.87
Suizhong 36-1	Offshore	0.003	0.98
Fengcheng	Onshore	0.65	8.15

Supplementary Table 6 CH₄ emission factors for different gas production fields

Field name	Field type	Exploration & Production (g m ⁻³)	Processing (g m ⁻³)
Shuangyushi&Jiulongshang	Onshore conventional	0.20	0.97
Anyue	Onshore conventional	0.03	1.05
Datianchi	Onshore conventional	1.06	0.80
Wolonghe	Onshore conventional	0.78	1.19
Mahe	Onshore conventional	1.28	0.89
Kelameili	Onshore conventional	1.65	0.43
Qingshen	Onshore conventional	1.22	0.92
Zhongba	Onshore conventional	1.33	0.86

Sebei	Onshore conventional	1.31	0.82
Tainan	Onshore conventional	1.33	0.82
Kekeya	Onshore conventional	1.33	0.92
Dongping	Onshore conventional	1.23	0.83
Luojiashai	Onshore conventional	0.00	0.23
Dukouhe	Onshore conventional	1.54	0.87
Longgang	Onshore conventional	0.00	1.51
Tieshanpo	Onshore conventional	0.00	0.55
Hetianhe	Onshore conventional	1.57	1.13
Yuanba	Onshore conventional	4.07	18.19
Puguang	Onshore conventional	1.22	0.00
Dina2	Onshore conventional	2.58	3.97
Kela	Onshore conventional	0.97	0.86
Yingmai7	Onshore conventional	1.29	0.93
Tahe	Onshore conventional	1.02	0.09
Tazhong	Onshore conventional	1.09	0.01
Changling&Songnan	Onshore conventional	0.47	0.00
Sulige	Onshore tight gas	12.06	0.80
Guangan	Onshore tight gas	15.66	0.76
Yingtai	Onshore tight gas	14.75	0.09
Hechuan	Onshore tight gas	13.31	1.44
Yulin	Onshore tight gas	16.56	0.96
Zhaotong	Onshore tight gas	17.25	0.76
Daniudi	Onshore tight gas	16.72	0.79
Bajiaochang	Onshore tight gas	18.64	0.78
Changning&Weiyuan	Onshore shale gas	16.23	0.83
Wushenqi	Onshore tight gas	16.27	1.66
Jingbian	Onshore tight gas	20.06	0.00
Yanchang	Onshore shale gas	19.95	0.30
Mizhi	Onshore tight gas	19.01	0.78
Zizhou	Onshore tight gas	15.95	1.31
Shenmu	Onshore tight gas	20.89	0.75
Xinchang	Onshore tight gas	14.24	0.80
Luodai	Onshore tight gas	17.62	0.81
Fuling	Onshore shale gas	21.39	0.79
Dabei	Onshore tight gas	16.74	0.88
Keshen	Onshore tight gas	17.01	0.89
Juggar CBM	Coalbed methane	4.05	0.83
Qinshui CBM	Coalbed methane	6.71	0.79
Bishuixing CBM	Coalbed methane	7.08	0.83
Ordos CBM	Coalbed methane	10.48	0.91
Chunxiao	Offshore	0.25	0.74
Liwan	Offshore	1.31	0.71
Panyu	Offshore	1.59	0.63

Lingshui	Offshore	1.92	1.46
Bozhong	Offshore	1.80	0.21
Ya	Offshore	2.16	1.17
Wenchang	Offshore	0.45	0.86
Ledong	Offshore	0.00	0.00
Dongfang	Offshore	0.73	1.18
Qiongx	Offshore	14.76	0.76

560

Table. The geospatial databases for emission mapping in this study.

Emission segment	Emission source	Source type	Geospatial database	Scarpelli et al., 2020 and 2022^{2, 3}
Oil exploration & production	Oil and gas field	Field	Havard_OILGAS database (https://maps.princeton.edu/catalog/harvard-glb-oilgas)	
Oil transport	Oil transport pipeline	Line	Global Energy Monitor (GEM) (https://globalenergymonitor.org/)	
Oil refining	Oil refinery	Point	National Energy and Technology Laboratory's Global Oil & Gas Infrastructure (GOGI) geodatabase (https://edx.netl.doe.gov/dataset/global-oil-gas-features-database).	Single data source: GOGI database
Gas exploration & production & processing	Oil and gas field	Field	Havard_OILGAS database	
Gas transmission	Gas transmission pipeline	Line	GEM GOGI database	
Gas storage	Gas storage facility	Point	GOGI database	
Gas import/export	LNG terminal	Point	GEM	
Gas distribution	Gas distribution pipeline	Other	Gridded population map from WorldPop from 2000 to 2020 (www.worldpop.org) Gridded land use map from the National Tibetan Plateau Data Center in 1990, 1995, 2000, 2005, 2010, and 2015 map (https://data.tpdac.ac.cn/home)	Did not use land cover map

(3) Methodological novelty

Our methods for both emission calculation and spatial allocation were improved. Multi-scale activity data was obtained in this work (Supplementary Table 3) and was further

565 harmonized across pipeline, city, provincial, and national levels to ensure data
consistency. In this study, the national-level data from the government was prioritized
as a top-level total constraint, given that within China's statistical framework, the
national-level data is collected by state statistical agencies through rigorous cross-
validation of multiple data sources. In cases where discrepancies arose between
570 national-level data and smaller-scale data, such as provincial or city-level data, we
corrected the smaller-scale data to maintain consistency. For example, city-level
production was constrained within the production volume of the country or province.
The pipeline-level length was evaluated and reconstructed to well match the national
total pipeline length. A detailed description of the calibration approach is presented in
the Emission estimation in Methods.

575 We also refined allocation approaches for CH₄ emissions from both oil and gas fields
and urban distribution pipeline systems. By the investigation of typical oil and gas field
areas in China and the consideration of buffer zone shapes, the potential
overestimations of emissions from production fields were rectified. The distribution
pattern of gas distribution emissions was reshaped and improved after adding time-
580 varying urban land cover maps to time-varying population density maps, as urban
populations are the primary consumers of natural gas in China.

2. One unique aspect of this study is the presentation of a long time series from 1990 to
2020, which, as the title indicates, allows for the exploration of structural shifts in
585 China's oil and gas emissions. However, the manuscript currently lacks adequate
information on how temporal data is processed, though the spatial mapping information
is well presented. For example, some key infrastructure data might be unavailable in
early years of the period or not continuous over time (e.g., GOGI). If this is the case,
how is the temporal information harmonized? What are the assumptions in generating
590 long-term trends. Including a discussion and clarification on this would be beneficial
for readers.

Response:

This study covers 7 major types of oil and gas infrastructure, including LNG terminals,
oil transport pipelines, gas transmission pipelines, gas distribution pipelines, oil and gas
595 production fields, oil refineries, and gas storage facilities. The start and end years of
operation for the first 4 types of infrastructure are provided by different datasets used
in the study, and those for the last 3 types remain unknown. Below, we will explain our
approaches to constructing a long-term emission map for both.

The GEM dataset provides operational periods of LNG terminals, oil transport pipelines,
600 and gas transmission pipelines. We compiled this information and allocated CH₄
emissions for each year to the facilities that were operational during that period. The
annual total length of the operational pipelines (after calibration) aligns with the annual
total length recorded officially during 1990-2022, indicating that the operational
periods derived from GEM are reasonably accurate. Regarding gas distribution

605 pipelines, we gathered the annual pipeline length for 347 prefecture-level cities from
 1990 to 2022 from the China Urban Construction Statistical Yearbook. The total length
 of gas distribution pipelines across all prefecture-level cities in each province closely
 matches the data from the National Bureau of Statistics from 1990 to 2022. These 4
 types of facilities with specific start and end years of operation contributed to 28% of
 610 the total CH₄ emissions from the oil and gas systems in China in 2022.

For production fields, oil refineries, and gas storage facilities, we relied on information
 from the Havard_OILGAS and GOGI databases, which do not include activity status
 for each year. To assess the activity status of production fields, we assume it depends
 on whether the city or province where the field is located recorded production volumes
 615 during that year. If the production volume was zero, the fields were considered inactive
 and not emitting. Conversely, if the production volume was greater than zero, the fields
 were considered active. Based on this assumption, we observed a fluctuating increase
 in the number of active oil and gas fields from 1990 to 2022 (Supplementary Fig. 9a).
 CH₄ emissions from production fields accounted for 66% of total emissions in 2022.
 620 The start year of operation for oil refineries and gas storage facilities is also unknown.
 Since their CH₄ emissions contributed only 6% of the national total in 2022, we assume
 that all the 176 refineries and 307 gas storage facilities in China have remained
 operational during 1990-2022.

625 **Supplementary Fig. 9 The number (or pipeline length) of the oil and gas
 infrastructure in this study and their CH₄ emissions from 1990 to 2022.** Note that
 the oil transport pipelines, gas transmission pipelines, and gas distribution pipelines are
 presented by their annual total lengths, while other facilities are shown as their annual
 total numbers.

630 We have added how we generate a long-term emission map for the oil and gas facilities
 in this study in Line 477-480, Line 483-484, Line 489-491, Line 494-495, and Line
 500-503 as follows:

Line 477-480: “The start year of operation for 176 oil refineries and 307 gas storage
 facilities in China is unknown and their CH₄ emissions contributed only 6% of national
 635 totals in 2022, we suppose that their status has remained operational and physical
 addresses were fixed from 1990 to 2022 (Supplementary Fig. 9).”

Line 483-484: “Annual emissions from LNG terminals are mapped to the facilities that were operational during each respective year.”

640 Line 489-491: “We begin by using GEM’s pipeline dataset to identify oil transport pipelines that were operational before the end of 2022. Next, we compile the pipelines in service for each year from 1990 to 2022 according to their respective start and end years.”

Line 494-495: “Similar to oil pipelines, the geographical routes of the operating pipelines for each year are collected and the emissions are mapped to these pipelines.”

645 Line 500-503: “Due to the unavailability of the field activity status for each year, we assume the activity status depends on whether the city or province where the field is located recorded production volumes during that year. If the production volume was zero, the fields were considered inactive and not emitting. In contrast, if the production volume was greater than zero, the fields were considered active.”

650

3. Additionally, it has been empirically shown that emission factors (or emission intensity) from the oil and gas system have been decreasing over time due to technological and management improvements. This aspect is not discussed in the paper. It appears that the authors used fixed emission factors, but changes in EF can have a substantial impact over a long period, which occurs to me a weakness of the

655

Response:

This work is constrained to use fixed emission factors due to the absence of comprehensive and representative dynamic emission factors of the oil and gas systems in China. To our current knowledge, previous bottom-up inventories tend to use fixed emission factors for China’s oil and gas systems^{2, 3, 6, 9, 13, 38}. We suspect that the improvements in low-emitting technologies within the Chinese oil and gas industry have been somewhat limited, as the relevant management practices remain unclear and were not released until the 2020s. For instance, the “Emissions Standards for Atmospheric Pollutants in Onshore Oil and Gas Exploration and Production Industry” was released in 2021, primarily targeting VOCs and SO₂ rather than CH₄. It wasn’t until 2023 that China published its first top-level document for CH₄ emissions control, the “Methane Emission Control Action Plan”, which still did not include explicit requirements for the extent of implementation of emissions reduction technologies in the oil and gas sectors. CH₄ emission mitigation practices of the oil and gas systems are in the pilot stage in China⁵¹. Although oil and gas companies might attempt to control CH₄ emissions for economic benefits, much of this information remains undisclosed. Therefore, we assume the emission factors do not change with time and we used the best available emission factors at this stage.

660

665

670

Based on a comprehensive review of existing studies on emissions of China’s oil and gas industry, we incorporated local emission factors wherever possible (Supplementary Table 4). In cases where the local factors are not available, the Tier 1 emission factors

675

recommended by the IPCC were adopted instead. For upstream oil and gas emissions, emission factors corresponding to the conditions of provinces or cities in China were estimated based on the emission intensities of domestic oil fields in 2015⁴⁰ and gas fields in 2016⁹. A detailed explanation of the province/city-level emission factors estimation and assumption is shown in the **Emission estimation** in Methods. Our estimated mean gas-production-normalized CH₄ loss rate of 1.6% is broadly consistent with a recent satellite-derived estimate in China³⁹. While the fixed parameters, though relatively reasonable, may underestimate historical upstream emissions and overestimate upstream emissions after 2016, our estimates broadly agree with other existing inventories. Furthermore, since the adoption of CH₄ emission control practices in China's oil and gas sector may be limited before 2023, this potential error of using the fixed emission factors is likely minor in analyzing emission trends.

In the gas sector's midstream and downstream emissions, we applied IPCC Tier 1 emission factors, which consist of two sets. The higher set represents limited use of lower-emission technologies and practices, while the lower set represents extensive use of these methods. We took the average from both sets and conducted a sensitivity analysis to evaluate the impacts of directly using high and low IPCC EF values on emission estimates. The results indicate that, due to the integration of local emission factors in upstream emissions, the choice between the high or low IPCC Tier 1 emission factors affects the final CH₄ emissions estimate of this study by approximately 15%.

4. Moreover, I'd request the authors to clarify their plan for data sharing, which is information missing in the current manuscript. The study's value would be greatly enhanced by making the data available to the community.

Response:

According to your suggestion, we will publicly share the data through the figshare (<https://figshare.com/s/137eaa7d0f63f5d14e6a>) if the paper is accepted.

Minor comments

1. L138: The calibration of pipeline distribution is well done. It would be worth illustrating this work, for example, by comparing the raw and corrected data in a supplementary figure.

Response:

We have added a supplementary figure (Supplementary Fig. 8) in SI and more descriptions in Lines 451-454 to clarify the calibration between the raw pipeline length and the corrected pipeline length. Please note that the "raw length" refers to the length directly from GEM.

715 **Supplementary Fig. 8 Comparison of the raw pipeline length, the official pipeline**
length, and the corrected pipeline length. (a) compares the total raw length and the
officially reported total length from 1990 to 2022. (b) compares the official reported
720 lengths (x-axis) of 246 pipelines with their respective lengths before and after
reconstruction (y-axis). The blue circles refer to the comparison between the official
lengths and the raw lengths. The purple circles indicate the comparison between the
1990-2022 average adjusted lengths with the official lengths. The red circles highlight
the CNPC Sichuan & Chongqing Network, showing a significant deviation between its
raw length and the recorded data. Its corrected length aligns closely with the official
length with an error margin of only 4%.

725

2. Section 2.3.3: I am not sure if the “buffer zone” method is necessary. It seems GFEI has used well-level information, which should be sufficient for spatially allocating emissions.

Response:

730 To spatially allocate upstream CH₄ emissions from the oil and gas sectors in China,
GFEI used gridded data that reflects the total number of wells per grid cell. However,
this well-level information is not used in this study for 3 main reasons below. (1) The
gridded data only includes the total number of wells, not differentiating between the
number of oil wells and gas wells. Given the marked differences in upstream emissions
735 between oil and gas, uniformly distributing total national upstream emissions over all
active wells, as GFEI did, can introduce considerable errors and fails to effectively
identify emission hotspots. (2) GFEI’s well-level data, derived from the Rose database,
is inconsistent with field-level information from the same source. Specifically, the
coverage of the gridded data is larger than the production fields (seen in Figure below),
740 suggesting the inconsistency within the same data source. (3) The offshore wells
belonging to China are not identified. China’s CH₄ emissions from the Bohai Sea, East
China Sea, and South China Sea are not clearly presented in GFEI’s gridded inventory.

Figure Redacted

745 Figure. GFEI’s well-level and field information from the Rose database (<https://doi.org/10.18141/1427300>).

750 Therefore, this study distributed the upstream emissions based on the Havard_OILGAS database, which provides the locations of center points of the oil and gas fields and their respective field types. It allows us to separately allocate upstream CH₄ emissions from onshore oil, offshore oil, onshore gas, and offshore gas sources to their central locations of the corresponding type of producing fields. We also investigated the areas of typical oil and gas fields in China and created a buffer zone around each point location to mitigate the overestimations of emissions from production fields with the center-point-based emission allocation approach. A detailed discussion of the spatial allocation approach is presented in **Emission mapping** in Results.

755

3. Section 2.3.4: It is commendable that the urban-rural difference is accounted for here. Does the urban grid cell mask change over time, or is it fixed?

Response:

760 The urban grid cell mask changes over time. We have added brief explanations in Lines 530-532 as follows:

By integrating land use grid data in 1990, 1995, 2000, 2005, 2010, and 2015 with annual population grids from 2000 to 2022, a total of 23 urban population grids were formed (Supplementary Table 7).

765 **Supplementary Table 7 Years of population and land use map for urban population grid data.**

Population map (year)	Land use cover map (year)	Urban population map (version)	Emission allocation years
---------------------------	--------------------------------	---------------------------

2000	1990	v1	1990~1994
2000	1995	v2	1995~1999
2000~2004	2000	v3~v7	2000~2004
2005~2009	2005	v8~v12	2005~2009
2010~2014	2010	v13~v17	2010~2014
2015~2020	2015	v18~v23	2015~2022

4. L47-48: Briefly define what you mean by “segment” here. This is a central concept in the paper, but the meaning is vague in the current description.

Response:

770 The “segment” characterized in this paper corresponds to IPCC subcategory 1B2, representing each part of the oil and gas systems according to their specific types of activity (seen in Table below).

Table. Detailed description for different segments in this study.

Emission segment	Detailed description
Oil exploration	This segment includes fugitive emissions (such as leaks from equipment, venting, and flaring) occurring during all field operations before production, including activities like prospecting, exploratory drilling, well testing, and completions.
Oil production	This segment refers to fugitive emissions from oil production mainly from the oil wellhead before transport and on-site processing of crude oil (i.e. removing water and gases in crude oil).
Oil transport	This segment covers venting and leakage emissions from transporting marketable crude oil to refineries, mainly from pipelines, tanker trucks, and rail cars.
Oil refining	This segment includes fugitive emissions from petroleum refineries, which process crude oils into final refined products, such as fuels and lubricants.
Gas exploration	This segment includes fugitive emissions occurring during all field operations before production, including activities like prospecting, exploratory drilling, well testing, and completions.
Gas production	This segment refers to fugitive emissions from the gas wellhead through to the entry points of gas processing plants or the tie-in points in transmission systems and includes emissions from gathering and boosting stations as well as gathering pipelines.
Gas processing	This segment includes fugitive emissions from gas processing facilities where natural gas liquids and other components are removed to produce “pipeline quality” gas for the transmission system.
Gas transmission	This segment includes fugitive emissions from high-pressure, large-diameter pipelines transporting processed natural gas over long distances from fields or processing sites to distribution systems or industrial customers (e.g., power

	plants or chemical facilities).
Gas storage	This segment includes fugitive emissions from storage facilities along transmission pipelines.
Gas import/export	This segment includes fugitive emissions associated with LNG stations and import/export terminals.
Gas distribution	This segment covers fugitive emissions from distribution pipelines that receive gas from the transmission system at “city gate” stations and then deliver the gas through underground mains and service lines to individual end consumers.

775 5. L59: There "is a" lack.

Response: Corrected.

6. L63: didn't -> did not.

Response: Corrected.

780

7. L103-104: It may be useful to tabulate the emission factors (EF) used in the study. I also wonder if the study accounts for changes in EF over the long term.

785 **Response:** We have added a table of emission factors as suggested (seen in Supplementary Table 4 above). The emission intensities for 14 oil fields and 59 gas fields in China are also provided (Supplementary Tables 5 and 6). The emission factors used in the study are fixed over time. Please refer to our response to the third general comment above for more explanations.

8. L236: surplus -> surpass.

790 **Response:** Corrected.

9. L238: flattening -> flat/stable.

Response: Corrected.

795 10. L224-223: This essentially repeats the last sentence.

Response: We have revised it for clarity and conciseness as follows:

800 Most of the increase occurred after 2000 with a total increase of 3.3 TgCH₄ from 2000 to 2022, which accounted for 93% of the total increase during the last three decades. The average annual growth rate after 2000 was 8% per year, double that of the 1990s (4%).

11. L314/L433: "reshaping" usually implies more drastic changes. In this case, it is a bit of an exaggeration, as the general pattern is similar to GFEI. I suggest that "improved" or "refined" better describes the changes.

805 **Response:** We have modified it as suggested.

Reference

1. Höglund-Isaksson L. Bottom-up simulations of methane and ethane emissions from global oil and gas systems 1980 to 2012. *Environ Res Lett* **12**, (2017).
- 810 2. Scarpelli TR, *et al.* A global gridded (0.1° × 0.1°) inventory of methane emissions from oil, gas, and coal exploitation based on national reports to the United Nations Framework Convention on Climate Change. *Earth System Science Data* **12**, 563-575 (2020).
3. Scarpelli TR, *et al.* Updated Global Fuel Exploitation Inventory (GFEI) for methane emissions from the oil, gas, and coal sectors: evaluation with inversions of atmospheric methane observations. *Atmospheric Chemistry and Physics* **22**, 3235-3249 (2022).
- 815 4. McDuffie EE, *et al.* A global anthropogenic emission inventory of atmospheric pollutants from sector- and fuel-specific sources (1970-2017): an application of the Community Emissions Data System (CEDS). *Earth System Science Data* **12**, 3413-3442 (2020).
5. O'Rourke PR, Smith, S. J., Mott, A., Ahsan, H., McDuffie, E. E., Crippa, M., Klimont, S., 820 McDonald, B., Z., Wang, Nicholson, M. B, Feng, L., and Hoesly, R. M. Community Emissions Data System (Version Feb-05-2021.) (2021).
6. Schwietzke S, Griffin WM, Matthews HS, Bruhwiler LMP. Global Bottom-Up Fossil Fuel Fugitive Methane and Ethane Emissions Inventory for Atmospheric Modeling. *ACS Sustainable Chemistry & Engineering* **2**, 1992-2001 (2014).
- 825 7. Janssens-Maenhout G, *et al.* EDGAR v4.3.2 Global Atlas of the three major greenhouse gas emissions for the period 1970-2012. *Earth System Science Data* **11**, 959-1002 (2019).
8. Global Non-CO₂ Greenhouse Gas Emission Projections & Mitigation Potential 2015-2050.). U.S. Environmental Protection Agency.
9. Gan Y, *et al.* Carbon footprint of global natural gas supplies to China. *Nature Communications* **11**, 9 (2020).
- 830 10. Southwest oil and gas Field Branch of petrochina Co. L. Natural gas pipeline transportation cost related information table.).
11. Buendia EC, Guendehou S, Limmeechokchai B, Pipatti R. 2019 Refinement to the 2006 IPCC Guidelines for National Greenhouse Gas Inventories.). Intergovernmental Panel on Climate Change (2019).
- 835 12. Peng S, *et al.* Inventory of anthropogenic methane emissions in mainland China from 1980 to 2010. *Atmospheric Chemistry and Physics* **16**, 14545-14562 (2016).
13. Liu G, *et al.* Recent Slowdown of Anthropogenic Methane Emissions in China Driven by Stabilized Coal Production. *Environmental Science & Technology Letters* **8**, 739-746 840 (2021).
14. *Dongying Statistical Yearbook*. China Statistics Press (2022).
15. *Jiangsu Economic Census Yearbook*. Jiangsu provincial Bureau of Statistics.
16. *Guangdong Economic Census Yearbook* Guangdong provincial Bureau of Statistics.
17. *Henan Economic Census Yearbook* Henan provincial Bureau of Statistics.
- 845 18. *China Land & Resources Almanac*. Ministry of Land and Resources.
19. *Yulin Statistical Yearbook*. China Statistics Press (2021).
20. *China Economic Yearbook*. Development Research Center of the State Council (2010).
21. *Jilin Economic Census Yearbook* Jilin provincial Bureau of Statistics.
22. *Inner Mongolia Economic Census Yearbook* Inner Mongolia provincial Bureau of Statistics.
- 850 23. *Shanxi Economic Census Yearbook* Shanxi provincial Bureau of Statistics.

24. *Sichuan Economic Census Yearbook*. Sichuan provincial Bureau of Statistics.
25. *China Statistical Yearbook*. State Statistics Bureau.
26. China Natural Gas Development Report.). Petrochina National high-end think tank research center.
- 855 27. Medium - and Long-Term Oil and Gas Pipeline Network Planning.). National Development and Reform Commission of China; National Energy Administration.
28. *China Urban Construction Statistical Yearbook*. Ministry of Housing and Urban-Rural Development of the People's Republic of China (MOHURD).
29. Chen G, *et al.* An improved method for estimating GHG emissions from onshore oil and gas exploration and development in China. *Sci Total Environ* **574**, 707-715 (2017).
- 860 30. Xue M, *et al.* Methane emissions from shale gas production sites in southern Sichuan, China: A field trial of methods. *Advances in Climate Change Research* **14**, 624-631 (2023).
31. Ge XH, *et al.* Monitoring and Investigating Methane Leakage in Coal Gas Production. *Pol J Environ Stud* **25**, 1005-1014 (2016).
- 865 32. Shuang Y, *et al.* Crude Oil Greenhouse Gas Emissions Characteristics and Influencing Factors in Shengtuo Oilfield. *Research of Environmental Sciences* **29**, 978-984 (2016).
33. Zhong Ja, Chen G, Zhang Z, Yang W, Wang Z. Greenhouse Gas Emissions from Natural Gas Development Process in a Gas Field in Sichuan Basin. *Research of Environmental Sciences* **28**, 355-360 (2015).
- 870 34. Chen C, Lv Y, He G. Estimating methane fugitive emissions from oil and natural gas systems in China. *Environmental Science* **43**, 4905-4913 (2022).
35. Yang Z, Gao J, Tang X, Zhong B, Zhang B. Accounting and spatial-temporal characteristics of fugitive methane emissions from the oil and natural gas industry in China. *Pet Sci Bull* **2**, 302-314 (2021).
- 875 36. Li X, *et al.* Estimation and reduction analysis of methane emissions in China's oil and gas industry. *Natural Gas Industry* **44**, (2024).
37. Zhang B, Chen GQ. China's CH₄ and CO₂ emissions: Bottom-up estimation and comparative analysis. *Ecological Indicators* **47**, 112-122 (2014).
38. Zhang B, Chen GQ, Li JS, Tao L. Methane emissions of energy activities in China 1980-
- 880 2007. *Renew Sust Energy Rev* **29**, 11-21 (2014).
39. Zhang YZ, *et al.* Observed changes in China's methane emissions linked to policy drivers. *Proc Natl Acad Sci U S A* **119**, 7 (2022).
40. Masnadi MS, *et al.* Well-to-refinery emissions and net-energy analysis of China's crude-oil supply. *Nat Energy* **3**, 220-226 (2018).
- 885 41. *National greenhouse gas inventory*. China Environment Press.
42. Crippa M, Guizzardi, D., Pagani, F., Banja, M., Muntean, M., Schaaf E., Becker, W., Monforti-Ferrario, F., Quadrelli, R., Riskey Martin, A., Taghavi-Moharamli, P., Köykkä, J., Grassi, G., Rossi, S., Brandao De Melo, J., Oom, D., Branco, A., San-Miguel, J., Vignati, E. In: *EDGAR (Emissions Database for Global Atmospheric Research) Community GHG Database*. version 8.0 edn (2023).
- 890 43. IEA. Global Methane Tracker 2023, IEA, Paris <https://www.iea.org/reports/global-methane-tracker-2023>, Licence: CC BY 4.0.) (2023).
44. Third Biennial Update Report on Climate Change of the People's Republic of China.). Ministry of Ecology and Environment of China (2023).

- 895 45. Saunois M, *et al.* The Global Methane Budget 2000–2017. *Earth System Science Data* **12**, 1561–1623 (2020).
46. Lauvaux T, *et al.* Global assessment of oil and gas methane ultra-emitters. *Science* **375**, 557–+ (2022).
47. Stavert AR, *et al.* Regional trends and drivers of the global methane budget. *Global Change Biology* **28**, 182–200 (2021).
- 900 48. , (!!! INVALID CITATION !!!).
49. Chen Z, *et al.* Methane emissions from China: a high-resolution inversion of TROPOMI satellite observations. *Atmospheric Chemistry and Physics* **22**, 10809–10826 (2022).
50. Lu X, *et al.* Global methane budget and trend, 2010–2017: complementarity of inverse analyses using in situ (GLOBALVIEWplus CH₄ ObsPack) and satellite (GOSAT) observations. *Atmospheric Chemistry and Physics* **21**, 4637–4657 (2021).
- 905 51. Yang X, Gao YY, Zhu MZ, Springer C. Assessing Methane Emissions From the Natural Gas Industry: Reviewing the Case of China in a Comparative Framework. *Curr Clim Chang Rep* **8**, 115–124 (2022).

910

Reviewer: 1

The manuscript construce a methane emission database of China's oil and gas system from 1990-2022 using the bottom-up method. Compared with previous global inventory studies, this study extends the time range and harmonizes multi-source data.

5 In the last round of responses, in view of the novelty in research, they gave a detailed reply in terms of the research boundary, data source, estimation method, and policy insights for emissions monitoring of the study. In the long time series of oil and gas methane emissions, it is good to see some novelty phenomena, such as the unexpected significant increase in the Shaanxi natural gas system. Although I mostly think the novelty phenomena can be found directly from activity data and emission factors. I have
10 a few comments for authors to address.

Response:

We express our gratitude to the referee for providing positive and constructive feedback on our manuscript. Below, we offer detailed responses addressing each point raised.

15

1. I previously mentioned that this manuscript is more suitable as a data-oriented study. Objectively speaking, the authors have made every effort to collect multi-source data and various emission factors to analyze the correlation between policy implementation and methane emissions from the oil and gas sector. Therefore, the authors emphasized
20 the significance of this work for national policy-making. I partially agree with the manuscript's viewpoint but feel that the importance has been somewhat overstated.

Response:

According to the referee's suggestion, we have made adjustments to the tone of certain statements in the text to provide a more balanced statement of our work's significance. Specifically, we revised the following lines:
25

Line 25-26: "This newly developed long-time series and spatially explicit CH₄ emission inventory contributes to informed policy decisions and swift climate action."

Line 76-77: "The emission dataset we develop can, to some extent, serve as a reference for CH₄ mitigation efforts."

30

2. The methane emissions from natural gas in Shaanxi Province (Supplementary Figure 2) were indeed unexpected. In the manuscript, Lines 314-316 mention that this is due to high emission factors. Previously, my understanding was that Sichuan and Xinjiang are the main natural gas production areas. Do other datasets, such as EDGARv8 and GFEL, also show high values for Shaanxi? Could you try to explain the reasons behind
35 the high emission factors in Shaanxi?

Response:

Previous inventories, including EDGAR and GFEI, did not identify high emissions from Shaanxi Province. This oversight is probably because they estimate emission trends relying on country-level gas production volumes and emission factors, which
40 constrains their ability to reveal distinct emission characteristics of different provinces.

The high emission factors in Shaanxi province are primarily due to its substantial tight gas production. Shaanxi encompasses China's major tight gas-producing fields, such as Sulige, Daniudi, and Jingbian, all of which together contribute to more than 90% of the province's total gas production. Tight gas is contained in dense rock formations with
45 small pore spaces and low permeability, making its extraction more challenging and categorizing it as unconventional natural gas. Compared with conventional gas, extracting tight gas from low-permeability formations requires energy-intensive processes, such as horizontal drilling and hydraulic fracturing, which lead to additional
50 emissions. Moreover, during the extraction process, there is a longer flow-back period associated with well completion and workover, resulting in fugitive emissions¹. Consequently, the CH₄ emission intensity from tight gas fields can be markedly higher than that from conventional gas fields, sometimes up to ten times larger. Our study has carefully considered these influential factors when estimating the specific emission
55 factors for Shaanxi Province, which explains the observed high emission factors.

We have added brief explanations in Supplementary Discussion 1 as follows:

Compared with conventional gas, extracting tight gas from low-permeability formations requires energy-intensive processes, such as horizontal drilling and hydraulic fracturing, which lead to additional emissions. Moreover, during the
60 extraction process, there is a longer flow-back period associated with well completion and workover, resulting in fugitive emissions.

Reviewer: 2

1. My main concern remains that this long-term study employs fixed emission factors.
65 The authors argue that there have not been dedicated methane reduction efforts in
China's oil and gas sector, thus justifying the assumption of a fixed EF. This argument,
however, is flawed. Emission factors can improve through advancements in technology
and better management practices, even if these improvements are not specifically aimed
70 at reducing methane emissions. For example, the recovery of associated gas for its
economic value can lead to lower emissions.

Empirical global analysis has shown that the average fossil-fuel fugitive emission rate
decreased from 7.6% in 1985 to 2.2% in 2013 (Schwietzke et al., 2016). While this FER
is not equivalent to EFs used in this study, this shows a long-term improvement in
efficiency. If China has followed this global trend, the current study, which uses fixed
75 EFs, may have overestimated the increase in emissions from China's oil and gas sector
by a factor of three. This has significant implications for the perceived importance of
the sector's emissions. While I understand that obtaining local, time-resolved, segment-
specific EFs is challenging, not acknowledging this limitation is problematic for a long-
term study.

80 **Response:**

We extend our appreciation to the referee for valuable feedback on our manuscript,
which has greatly assisted our research. Below, we address each comment in detail.

We acknowledge that employing fixed emission factors has limitations, particularly due
to the challenges of obtaining local, time-resolved, segment-specific emission factors
85 for China's oil and gas systems. To our current knowledge, previous bottom-up
inventories tend to use fixed emission factors for China's oil and gas systems¹⁻⁷. Our
estimates broadly agree with other existing emission inventories in national totals.

To further address the concerns, we conducted a sensitivity analysis to discuss the
limitations and uncertainties associated with the emission factors. The GAINS model
90 projected a yearly enhancement of 1% in the reduction efficiency (RE) for non-CO₂
greenhouse gas (GHG) reduction technologies. Similarly, the USEPA estimated a 1%
annual enhancement for emerging non-CO₂ GHG reduction technologies and chose a
more cautious rate of 0.5% per year for mature technologies⁸. Given that China is
classified as an emerging economy, we suspect that the abatement technologies in China
95 are not as mature as those in developed regions. Based on the assumptions outlined in
the GAINS and USEPA frameworks for non-CO₂ emissions, we have assumed that
between 2000 and 2022, mitigation measures in China led to a 1% annual reduction in
upstream emission factors for oil and gas systems. For midstream and downstream
emissions, such as pipeline leaks, there have been no corresponding reduction policies
100 implemented, and related technologies remain underutilized. For the period from 1990
to 2000, we did not consider any improvements in emission reduction technologies.

The sensitivity results indicate that our emission trends exhibit robustness, with national

emission variations not exceeding 16% from 2000 to 2022 (Supplementary Fig. 7). Our findings reveal a relatively small impact of time-varying emission factors on estimation trends, at least on an annual national scale, with emission variations mostly less than 12%.

Supplementary Fig. 7 Comparison of annual CH₄ emissions relative to the average in China under different experiments and studies. This study (time-varying-EF) indicates the estimates using the upstream emission factors with a 1% annual reduction. Details of uncertainty discussion and evaluation of emission estimates have been added in the Uncertainties and limitations in Results.

2. L173-175: May need to introduce what "town gas" is. I am not familiar with the concept. How do town gas's composition and emission factors differ from pipeline natural gas?

Response:

Town gas, also known as city gas or coal gas, is a manufactured gaseous fuel produced primarily from the gasification of coal. It serves as an energy source in both commercial and residential applications. The composition of town gas typically includes a mixture of hydrogen (H₂), methane (CH₄), and other gases such as carbon monoxide (CO), carbon dioxide (CO₂), and nitrogen (N₂). According to the IPCC guidelines, the estimated typical composition of town gas is approximately 50% H₂, 35% CH₄, 10% CO, and around 5% volatile hydrocarbons, along with trace amounts of CO₂ and N₂⁹. In contrast, pipeline natural gas is predominantly composed of CH₄, with a much lower proportion of other components. This difference in composition leads to different emission factors for town gas compared to natural gas. The emission factor for the natural gas distribution segment is estimated to be between 0.23 and 1.17 kg m⁻¹, while the emission factor for town gas distribution is approximately 0.58 kg m⁻¹, as per the IPCC guidelines.

We have added brief explanations in Lines 172-176 as follows:

From 1990 to 2000, CH₄ emissions from gas distribution in Chinese cities were dominated by town gas, a manufactured gaseous fuel produced mainly through the gasification of coal. The composition of town gas differs from that of natural gas, with hydrogen (H₂) being the predominant component, followed by CH₄⁹.

3. L499-500: the URL is not accessible.

Response: Corrected.

140 **Reference**

1. Gan Y, *et al.* Carbon footprint of global natural gas supplies to China. *Nature Communications* **11**, 9 (2020).
2. Scarpelli TR, *et al.* A global gridded (0.1° × 0.1°) inventory of methane emissions from oil, gas, and coal exploitation based on national reports to the United Nations Framework
145 Convention on Climate Change. *Earth System Science Data* **12**, 563-575 (2020).
3. Scarpelli TR, *et al.* Updated Global Fuel Exploitation Inventory (GFEI) for methane emissions from the oil, gas, and coal sectors: evaluation with inversions of atmospheric methane observations. *Atmospheric Chemistry and Physics* **22**, 3235-3249 (2022).
4. Liu G, *et al.* Recent Slowdown of Anthropogenic Methane Emissions in China Driven by
150 Stabilized Coal Production. *Environmental Science & Technology Letters* **8**, 739-746 (2021).
5. Zhang B, Chen GQ, Li JS, Tao L. Methane emissions of energy activities in China 1980-2007. *Renew Sust Energy Rev* **29**, 11-21 (2014).
6. Li X, *et al.* Estimation and reduction analysis of methane emissions in China's oil and gas
155 industry. *Natural Gas Industry* **44**, (2024).
7. Yang Z, Gao J, Tang X, Zhong B, Zhang B. Accounting and spatial-temporal characteristics of fugitive methane emissions from the oil and natural gas industry in China. *Pet Sci Bull* **2**, 302-314 (2021).
8. Global Non-CO2 Greenhouse Gas Emissions Projections & Marginal Abatement Cost Analysis.). U.S. Environmental Protection Agency (2019).
160
9. Buendia EC, Guendehou S, Limmeechokchai B, Pipatti R. 2019 Refinement to the 2006 IPCC Guidelines for National Greenhouse Gas Inventories.). Intergovernmental Panel on Climate Change (2019).

Reviewer: 2

The authors conducted a sensitivity analysis to evaluate the uncertainty associated with time-varying emission factors, and the findings are presented in Fig. S7. If I understand correctly, Fig. S7 compares the results of "This study (IPCC-EF)" and "This study (time-varying-EF)," and time-varying emission factors are expected to yield lower results compared to fixed emission factors. But based on the figure, the difference between the two (almost overlap) seems smaller than the up to 16% difference mentioned in the text. Moreover, the results for time-varying-EF even appear slightly higher in the figure. Please double-check the accuracy of this figure. Other than this, I have no further comments. The authors have addressed reviewers' comments.

Response:

Thank you for your feedback.

Fig. S7 compares the results of "This study" (red line) and "This study (time-varying-EF)," not "This study (IPCC-EF)" and "This study (time-varying-EF)." "This study (IPCC-EF)" refers to estimates using the same method and activity data but applying IPCC emission factors. Time-varying emission factors yield lower results compared to fixed local emission factors (red line). The 16% difference mentioned in the text refers to national emission variations between "This study" and "This study (time-varying-EF)," not between "This study (IPCC-EF)" and "This study (time-varying-EF)."

20

Supplementary Fig. 7 Comparison of annual CH₄ emissions relative to the average in China under different experiments and studies. This study indicates the estimates applying the local emission factors. This study (IPCC-EF) indicates the estimates using the same method and activity data but applying the IPCC emission factors. This study (time-varying-EF) indicates the estimates using the local upstream emission factors with a 1% annual reduction.

25